# Associations of autozygosity with a broad range of human phenotypes

David W Clark [ID] et al.[#]

In many species, the offspring of related parents suffer reduced reproductive success, a phenomenon known as inbreeding depression. In humans, the importance of this effect has remained unclear, partly because reproduction between close relatives is both rare and frequently associated with confounding social factors. Here, using genomic inbreeding coefficients ($F_{ROH}$) for >1.4 million individuals, we show that $F_{ROH}$ is significantly associated ($p < 0.0005$) with apparently deleterious changes in 32 out of 100 traits analysed. These changes are associated with runs of homozygosity (ROH), but not with common variant homozygosity, suggesting that genetic variants associated with inbreeding depression are predominantly rare. The effect on fertility is striking: $F_{ROH}$ equivalent to the offspring of first cousins is associated with a 55% decrease [95% CI 44–66%] in the odds of having children. Finally, the effects of $F_{ROH}$ are confirmed within full-sibling pairs, where the variation in $F_{ROH}$ is independent of all environmental confounding.

*email: jim.wilson@ed.ac.uk. [#]A full list of authors and their affiliations appears at the end of the paper.

Given the pervasive impact of purifying selection on all populations, it is expected that genetic variants with large deleterious effects on evolutionary fitness will be both rare and recessive[1]. However, precisely because they are rare, most of these variants have yet to be identified and their recessive impact on the global burden of disease is poorly understood. This is of particular importance for the nearly one billion people living in populations where consanguineous marriages are common[2], and the burden of genetic disease is thought to be disproportionately due to increased homozygosity of rare, recessive variants[3–5]. Although individual recessive variants are difficult to identify, the net directional effect of all recessive variants on phenotypes can be quantified by studying the effect of inbreeding[6], which gives rise to autozygosity (homozygosity due to inheritance of an allele identical-by-descent).

Levels of autozygosity are low in most of the cohorts with genome-wide data[7,8] and consequently very large samples are required to study the phenotypic impact of inbreeding[9]. Here, we meta-analyse results from 119 independent cohorts to quantify the effect of inbreeding on 45 commonly measured complex traits of biomedical or evolutionary importance, and supplement these with analysis of 55 more rarely measured traits included in UK Biobank[10].

Continuous segments of homozygous alleles, or runs of homozygosity (ROH), arise when identical-by-descent haplotypes are inherited down both sides of a family. The fraction of each autosomal genome in ROH > 1.5 Mb ($F_{ROH}$) correlates well with pedigree-based estimates of inbreeding[11]. We estimate $F_{ROH}$ using standard methods and software[6,12] for a total of 1,401,776 individuals in 234 uniform sub-cohorts. The traits measured in each cohort vary according to original study purpose, but together cover a comprehensive range of human phenotypes (Fig. 1, Supplementary Data 7). The five most frequently contributed traits (height, weight, body mass index, systolic and diastolic blood pressure) are measured in >1,000,000 individuals; a further 16 traits are measured >500,000 times.

We find that $F_{ROH}$ is significantly associated with apparently deleterious changes in 32 out of 100 traits analysed. Increased $F_{ROH}$ is associated with reduced reproductive success (decreased number and likelihood of having children, older age at first sex and first birth, decreased number of sexual partners), as well as reduced risk-taking behaviour (alcohol intake, ever-smoked, self-reported risk taking) and increased disease risk (self-reported overall health and risk factors including grip strength and heart rate). We show that the observed effects are predominantly associated with rare (not common) variants and, for a subset of traits, differ between men and women. Finally, we introduce a within-siblings method, which confirms that social confounding of $F_{ROH}$ is modest for most traits. We therefore conclude that inbreeding depression influences a broad range of human phenotypes through the action of rare, recessive variants.

## Results

**Cohort characteristics**. As expected, cohorts with different demographic histories varied widely in mean $F_{ROH}$. The within-cohort standard deviation of $F_{ROH}$ is strongly correlated with the mean (Pearson's $r = 0.82$; Supplementary Fig. 3), and the most homozygous cohorts provide up to 100 times greater per-sample statistical power than cosmopolitan European-ancestry cohorts (Supplementary Data 5). To categorise cohorts, we plotted mean $F_{ROH}$ against $F_{IS}$ (Fig. 2). $F_{IS}$ measures inbreeding as reflected by non-random mating in the most recent generation, and is calculated as the mean individual departure from Hardy–Weinberg equilibrium ($F_{SNP}$; see Methods). Cohorts with high rates of consanguinity lie near the $F_{ROH} = F_{IS}$ line, since most excess SNP homozygosity is caused by ROH. In contrast, cohorts with small effective population sizes, such as the Amish and Hutterite isolates of North America, have high average $F_{ROH}$, often despite avoidance of mating with known relatives, since identical-by-descent haplotypes are carried by many couples, due to a restricted number of possible ancestors.

**Traits affected by $F_{ROH}$**. To estimate the effect of inbreeding on each of the 100 phenotypes studied, trait values were regressed on $F_{ROH}$ within each cohort, taking account of covariates including

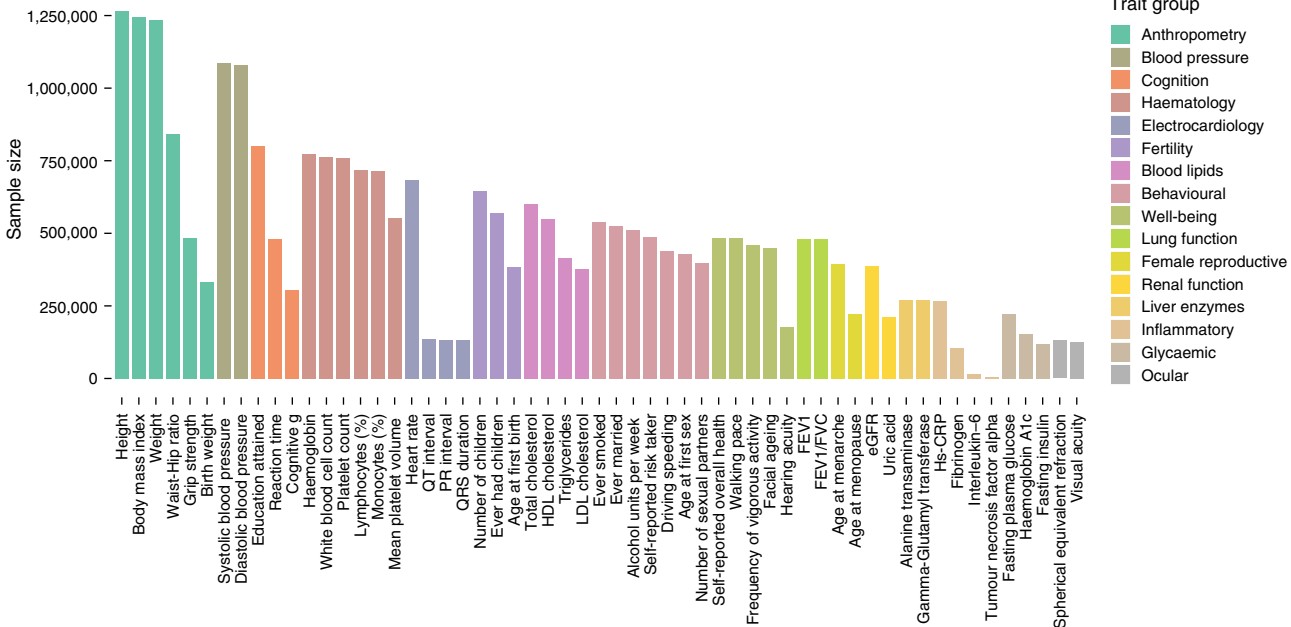

**Fig. 1** Census of complex traits. Sample sizes are given for analyses of 57 representative phenotypes, arranged into 16 groups covering major organ systems and disease risk factors. HDL high-density lipoprotein, LDL low-density lipoprotein, hs-CRP high-sensitivity C-reactive protein, TNF-alpha tumour necrosis factor alpha, FEV1 forced expiratory volume in one second, FVC forced vital capacity, eGFR estimated glomerular filtration rate

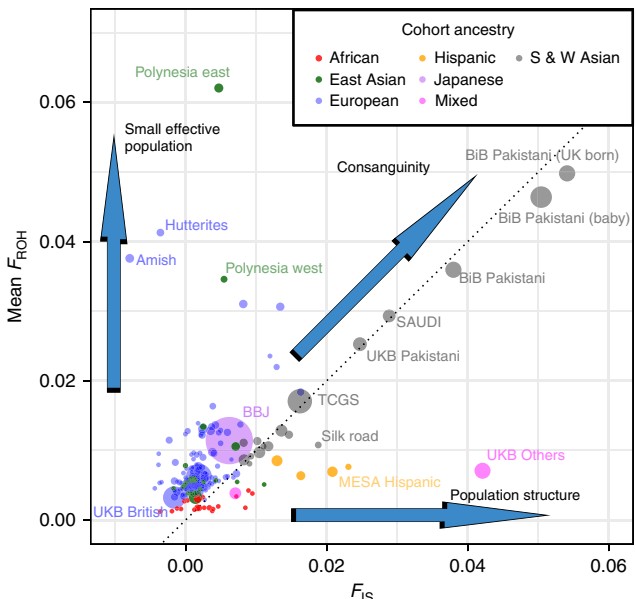

**Fig. 2** Mean $F_{ROH}$ and $F_{IS}$ for 234 ROHgen sub-cohorts. Each cohort is represented by a circle whose area is proportional to the approximate statistical power ($N\sigma^2_{F_{ROH}}$) contributed to estimates of $\beta_{F_{ROH}}$. Mean $F_{ROH}$ can be considered as an estimate of total inbreeding relative to an unknown base generation, approximately tens of generations past. $F_{IS}$ measures inbreeding in the current generation, with $F_{IS} = 0$ indicating random mating, $F_{IS} > 0$ indicating consanguinity, and $F_{IS} < 0$ inbreeding avoidance[46]. In cohorts along the y-axis, such as the Polynesians and the Anabaptist isolates, autozygosity is primarily caused by small effective population size rather than preferential consanguineous unions. In contrast, in cohorts along the dotted unity line, all excess SNP homozygosity is accounted for by ROH, as expected of consanguinity within a large effective population. A small number of cohorts along the x-axis, such as Hispanic and mixed-race groups, show excess SNP homozygosity without elevated mean $F_{ROH}$, indicating population genetic structuring, caused for instance by admixture and known as the Wahlund effect. A few notable cohorts are labelled. BBJ Biobank Japan, BiB Born in Bradford, UKB UK Biobank, MESA Multiethnic Study of Atherosclerosis, TCGS Tehran Cardiometabolic Genetic Study

age, sex, principal components of ancestry and, in family studies, a genomic relationship matrix (GRM) (Supplementary Data 3). Cross-cohort effect size estimates were then obtained by fixed-effect, inverse variance-weighted meta-analysis of the within-cohort estimates (Supplementary Data 10). Twenty-seven out of 79 quantitative traits and 5 out of 21 binary traits reach experiment-wise significance (0.05/100 or $p < 0.0005$; Fig. 3a, b). Among these are replications of the previously reported effects on reduction in height[13], forced expiratory lung volume in one second, cognition and education attained[6]. We find that the 32 phenotypes affected by inbreeding can be grouped into five broader categories: reproductive success, risky behaviours, cognitive ability, body size, and health.

Despite the greater individual control over reproduction in the modern era, due to contraception and fertility treatments, we find that increased $F_{ROH}$ has significant negative effects on five traits closely related to fertility. For example, an increase of 0.0625 in $F_{ROH}$ (equivalent to the difference between the offspring of first cousins and those of unrelated parents) is associated with having 0.10 fewer children [$\beta_{0.0625} = -0.10 \pm 0.03$ 95% confidence interval (CI), $p = 1.8 \times 10^{-10}$]. This effect is due to increased $F_{ROH}$ being associated with reduced odds of having any children (OR$_{0.0625} = 0.65 \pm 0.04$, $p = 1.7 \times 10^{-32}$) as opposed to fewer children among parents ($\beta_{0.0625} = 0.007 \pm 0.03$, $p = 0.66$). Since

autozygosity also decreases the likelihood of having children in the subset of individuals who are, or have been, married, (OR$_{0.0625} = 0.71 \pm 0.09$, $p = 3.8 \times 10^{-8}$) it appears that the cause is a reduced ability or desire to have children, rather than reduced opportunity. Consistent with this interpretation, we observe no significant effect on the likelihood of marriage (OR$_{0.0625} = 0.94 \pm 0.07$, $p = 0.12$) (Fig. 3b). All effect size, odds ratios and 95% CI are stated as the difference between $F_{ROH} = 0$ and $F_{ROH} = 0.0625$.

The effects on fertility may be partly explained by the effect of $F_{ROH}$ on a second group of traits, which capture risky or addictive behaviour. Increased $F_{ROH}$ is associated with later age at first sex ($\beta_{0.0625} = 0.83 \pm 0.19$ years, $p = 5.8 \times 10^{-17}$) and fewer sexual partners ($\beta_{0.0625} = -1.38 \pm 0.38$, $p = 2.0 \times 10^{-12}$) but also reduced alcohol consumption ($\beta_{0.0625} = -0.66 \pm 0.12$ units per week, $p = 1.3 \times 10^{-22}$), decreased likelihood of smoking (OR$_{0.0625} = 0.79 \pm 0.05$, $p = 5.9 \times 10^{-13}$), and a lower probability of being a self-declared risk-taker (OR$_{0.0625} = 0.84 \pm 0.06$, $p = 3.4 \times 10^{-5}$) or exceeding the speed limit on a motorway ($p = 4.0 \times 10^{-8}$). Conservative beliefs are likely to affect these traits, and are known to be confounded with $F_{ROH}$ in some populations[14], however, fitting religious participation as a covariate in UKB reduces, but does not eliminate the reported effects (Supplementary Fig. 10b, Supplementary Data 20). Similarly, fitting educational attainment as an additional covariate reduces 16 of 25 significant effect estimates, but actually increases 9, including age at first sex and number of children (Supplementary Fig. 10a, Supplementary Data 20). This is because reduced educational attainment is associated with earlier age at first sex and increased number of children, which makes it an unlikely confounder for the effects of $F_{ROH}$, which are in the opposite directions.

A third group of traits relates to cognitive ability. As previously reported, increased autozygosity is associated with decreased general cognitive ability, $g$[6,15] and reduced educational attainment[6]. Here, we also observe an increase in reaction time ($\beta_{0.0625} = 11.6 \pm 3.9$ ms, $p = 6.5 \times 10^{-9}$), a correlate of general cognitive ability (Fig. 3a, Supplementary Data 10).

A fourth group relates to body size. We replicate previously reported decreases in height and forced expiratory volume[6] (Supplementary Data 21) and we find that increased $F_{ROH}$ is correlated with a reduction in weight ($\beta_{0.0625} = 0.86 \pm 0.12$ kg, $p = 3.4 \times 10^{-28}$) and an increase in the waist to hip ratio ($\beta_{0.0625} = 0.004 \pm 0.001$, $p = 1.4 \times 10^{-11}$).

The remaining effects are loosely related to health and frailty; higher $F_{ROH}$ individuals report significantly lower overall health and slower walking pace, have reduced grip strength ($\beta_{0.0625} = -1.24 \pm 0.19$ kg, $p = 6.9 \times 10^{-24}$), accelerated self-reported facial ageing, and poorer eyesight and hearing. Increased $F_{ROH}$ is also associated with faster heart rate ($\beta_{0.0625} = 0.56 \pm 0.24$ bpm, $p = 5.9 \times 10^{-6}$), lower haemoglobin ($\beta_{0.0625} = 0.81 \pm 0.24$ gL$^{-1}$, $p = 1.6 \times 10^{-11}$), lymphocyte percentage, and total cholesterol ($\beta_{0.0625} = -0.05 \pm 0.015$ mmol L$^{-1}$, $p = 5.2 \times 10^{-10}$).

**Sex-specific effects of $F_{ROH}$.** Intriguingly, for a minority of traits (13/100), the effect of $F_{ROH}$ differs between men and women (Fig. 3c, Supplementary Data 12). For example, men who are the offspring of first cousins have 0.10 mmol L$^{-1}$ [95% CI 0.08–0.12] lower total cholesterol on average, while there is no significant effect in women; LDL shows a similar pattern. More generally, for these traits, the effect in men is often of greater magnitude than the effect in women, perhaps reflecting differing relationships between phenotype and fitness.

**Associations most likely caused by rare, recessive variants.** The use of ROH to estimate inbreeding coefficients is relatively new in inbreeding research[11,16–19]. Earlier frequency-based estimators

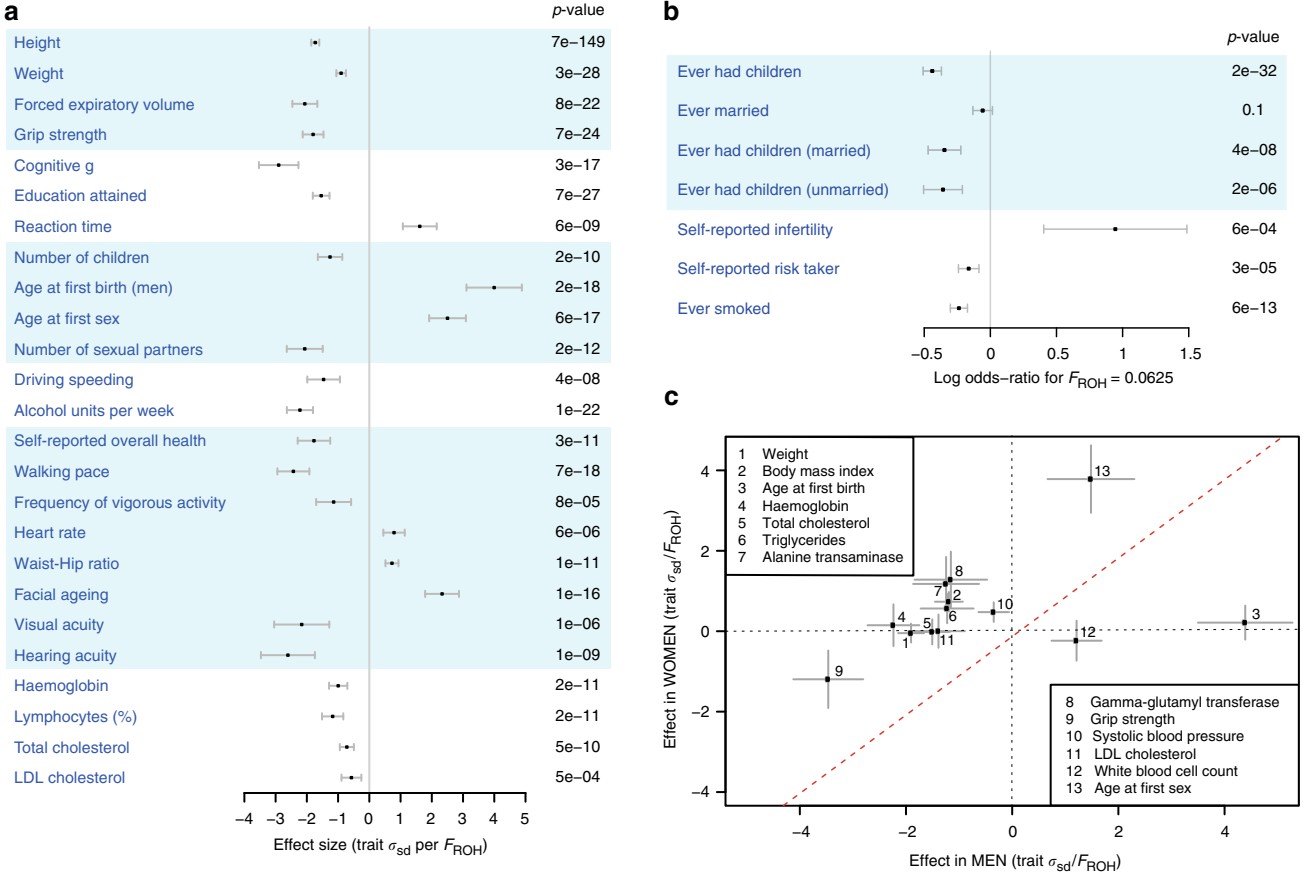

**Fig. 3** Scope of inbreeding depression. **a** Effect of $F_{ROH}$ on 25 quantitative traits. To facilitate comparison between traits, effect estimates are presented in units of within-sex standard deviations. Traits shown here reached Bonferroni-corrected significance of $p = 0.0005$ (=0.05/100 traits). Sample sizes, within-sex standard deviations, and effect estimates in measurement units are shown in Supplementary Data 9. FEV1 forced expiratory volume in one second. Traits are grouped by type. **b** Effect of $F_{ROH}$ on eight binary traits with associated $p$ values. Effect estimates are reported as ln(Odds-Ratio) for the offspring of first cousins, for which $E(F_{ROH}) = 0.0625$. Self-declared infertility is shown for information, although this trait does not reach Bonferroni corrected significant ($OR_{0.0625} = 2.6 \pm 1.1$, $p = 0.0006$). Numbers of cases and controls and effect estimates for all binary traits are shown in Supplementary Data 10. **c** Sex-specificity of ROH effects. The effect of $F_{ROH}$ in men versus that in women is shown for 13 traits for which there was evidence of significant differences in the effects between sexes. For 11 of these 13 traits the magnitude of effect is greater in men than in women. Traits such as liver enzymes levels (alanine transaminase, gamma-glutamyl transferase) show sex-specific effects of opposite sign (positive in women, negative in men), which cancel out in the overall analysis. BMI body mass index, LDL low-density lipoprotein. All errors bars represent 95% confidence intervals

such as $F_{SNP}$ and $F_{GRM}$[20], made use of excess marker homozygosity[21–23] and did not require physical maps. We performed both univariate and multivariate regressions to evaluate the effectiveness of $F_{ROH}$ against these measures. The correlations between them range from 0.13 to 0.99 and are strongest in cohorts with high average inbreeding (Supplementary Data 6, Supplementary Fig. 6). Significantly, univariate regressions of traits on both $F_{SNP}$ and $F_{GRM}$ show attenuated effect estimates relative to $F_{ROH}$ (Supplementary Data 13). This attenuation is greatest in low autozygosity cohorts, suggesting that $F_{ROH}$ is a better estimator of excess homozygosity at the causal loci (Fig. 4c).

To explore this further, we fit bivariate models with $F_{ROH}$ and $F_{GRM}$ as explanatory variables. For all 32 traits that were significant in the univariate analysis, we find that $\widehat{\beta}_{F_{ROH}|F_{GRM}}$ is of greater magnitude than $\widehat{\beta}_{F_{GRM}|F_{ROH}}$ in the conditional analysis (Fig. 4b, Supplementary Data 22). This suggests that inbreeding depression is predominantly caused by rare, recessive variants made homozygous in ROH, and not by the chance homozygosity of variants in strong LD with common SNPs (Fig. 4d, Supplementary Note 5). We also find that ROH of different

lengths have similar effects per unit length (Fig. 4a, Supplementary Fig. 11a), consistent with their having a causal effect on traits and not with confounding by socioeconomic or other factors, as shorter ROH arise from deep in the pedigree are thus less correlated with recent consanguinity.

**Quantifying the scope of social confounding.** Previous studies have highlighted the potential for $F_{ROH}$ to be confounded by non-genetic factors[6,24]. We therefore estimated the effect of $F_{ROH}$ within various groups, between which confounding might be expected either to differ, or not be present at all.

For example, the effect of $F_{ROH}$ on height is consistent across seven major continental ancestry groups (Supplementary Fig. 1, Supplementary Data 18), despite differing attitudes towards consanguinity, and consequently different burdens and origins of ROH. Similarly, grouping cohorts into consanguineous, more cosmopolitan, admixed and those with homozygosity due to ancient founder effects also shows consistent effects (Supplementary Fig. 2, Supplementary Data 19). Equally, categorising samples into bins of increasing $F_{ROH}$ shows a dose-dependent response of the study traits with increased $F_{ROH}$ (Supplementary Data 17 and

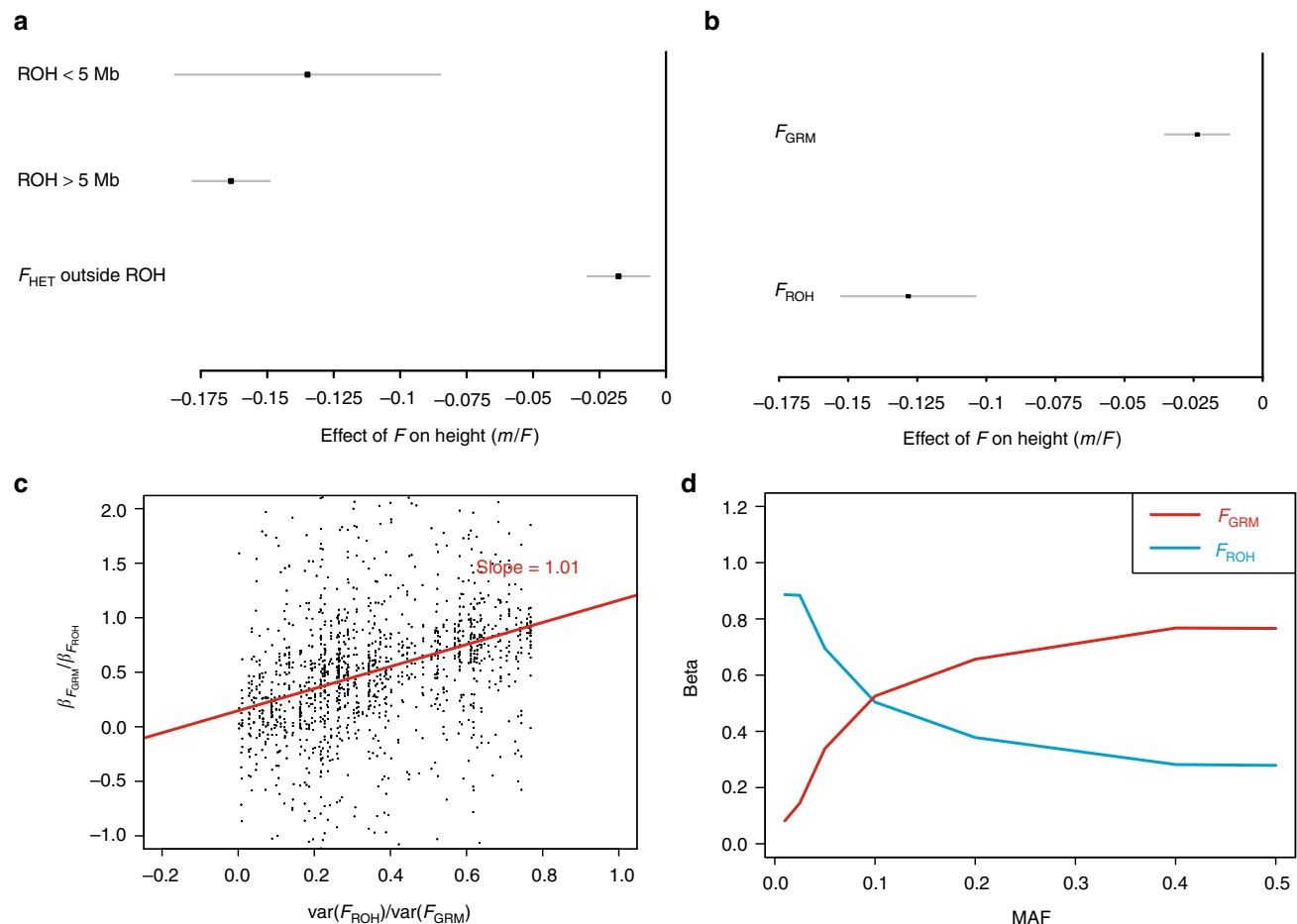

**Fig. 4** Inbreeding depression caused by ROH. **a** Effect of different ROH lengths on height, compared with the effect of SNP homozygosity outside of ROH. The effects of shorter (<5 Mb) and longer (>5 Mb) ROH per unit length are similar and strongly negative, whereas the effect of homozygosity outside ROH is much weaker. The pattern is similar for other traits (Supplementary Fig. 11a; Supplementary Data 14). **b** $F_{ROH}$ is more strongly associated than $F_{GRM}$ in a bivariate model of height. Meta-analysed effect estimates, and 95% confidence intervals, are shown for a bivariate model of height (Height $\sim F_{ROH} + F_{GRM}$). The reduction in height is more strongly associated with $F_{ROH}$ than $F_{GRM}$, as predicted if the causal variants are in weak LD with the common SNPs used to calculate $F_{GRM}$ (Supplementary Note 5). The pattern is similar for other traits (Supplementary Fig. 15a, b; Supplementary Data 22). **c** $F_{ROH}$ is a lower variance estimator of the inbreeding coefficient than $F_{GRM}$. The ratio of $\beta_{F_{GRM}} : \beta_{F_{ROH}}$ is plotted against $\frac{\mathrm{var}(F_{ROH})}{\mathrm{var}(F_{GRM})}$ for all traits in all cohorts. When the variation of $F_{GRM}$ which is independent of $F_{ROH}$ has no effect on traits, $\hat{\beta}_{F_{GRM}}$ is downwardly biased by a factor of $\frac{\mathrm{var}(F_{ROH})}{\mathrm{var}(F_{GRM})}$ (Supplementary Note 4). A linear maximum likelihood fit, shown in red, has a gradient consistent with unity [1.01; 95% CI 0.84–1.18], as expected when the difference between $F_{GRM}$ and $F_{ROH}$ is not informative about the excess homozygosity at causal variants (Supplementary Note 5). **d** $F_{ROH}$ is a better predictor of rare variant homozygosity than $F_{GRM}$. The excess homozygosities of SNPs, extracted from UK Biobank imputed genotypes, were calculated at seven discrete minor allele frequencies ($F_{MAF}$), and regressed on two estimators of inbreeding in a bivariate statistical model (see Supplementary Note 5). The homozygosity of common SNPs is better predicted by $F_{GRM}$, but rare variant homozygosity is better predicted by $F_{ROH}$. The results from real data (Fig. 4b, Supplementary Figs 15a, b and Supplementary Data 22) are consistent with those simulated here, if the causal variants are predominantly rare. All errors bars represent 95% confidence intervals

Fig. 5a, b show the response for height and ever having children; Supplementary Figs 9a–f for all significant traits). The proportionality of these effects is consistent with a genetic cause, while it is difficult to envisage a confounder proportionally associated across the *entire* range of observed $F_{ROH}$. In particular, the highest $F_{ROH}$ group ($F_{ROH} > 0.18$), equivalent to the offspring of first-degree relatives, are found to be, on average, 3.4 [95% CI 2.5–4.3] cm shorter and 3.1 [95% CI 2.5–3.7] times more likely to be childless than an $F_{ROH} = 0$ individual.

Next, we estimated $\beta_{F_{ROH}}$ for 7153 self-declared adopted individuals in UK Biobank, whose genotype is less likely to be confounded by cultural factors associated with the relatedness of their biological parents. For all 26 significant traits measured in this cohort, effect estimates are directionally consistent with the meta-analysis and 3 (height, walking pace and hearing acuity)

reach replication significance ($p < 0.004$). In addition, a meta-analysis of the ratio $\hat{\beta}_{F_{ROH\_ADOPTEE}} : \hat{\beta}_{F_{ROH}}$ across all traits differs significantly from zero (Fig. 5c; average = 0.78, 95% CI 0.56–1.00, $p = 2 \times 10^{-12}$).

Finally, the effect of $F_{ROH}$ was estimated in up to 118,773 individuals in sibships (full-sibling pairs, trios, etc.: $\hat{\beta}_{F_{ROH\_wSibs}}$). $F_{ROH}$ differences between siblings are caused entirely by Mendelian segregation, and are thus independent of any reasonable model of confounding. The variation of $F_{ROH}$ among siblings is a small fraction of the population-wide variation[11] (Supplementary Data 5); nevertheless, 23 out of 29 estimates of $\hat{\beta}_{F_{ROH\_wSibs}}$ are directionally consistent with $\hat{\beta}_{F_{ROH}}$, and two (self-reported overall health and ever having children) reach replication

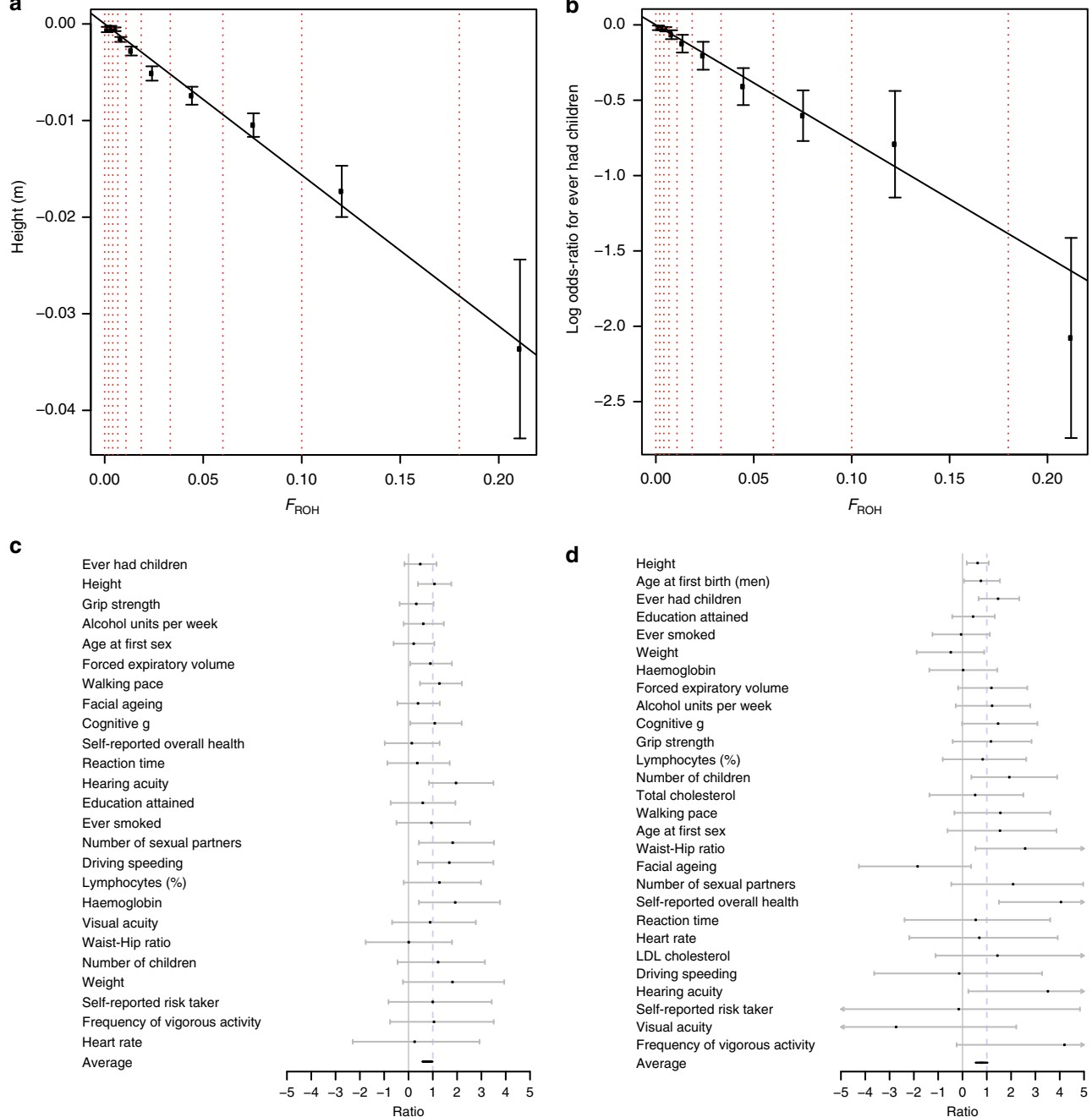

**Fig. 5** Evidence ROH effects are un-confounded. **a** Linear decrease in height with increasing $F_{ROH}$. Average heights (in metres) is plotted in bins of increasing $F_{ROH}$. The limits of each bin are shown by red dotted lines, and correspond to the offspring of increasing degree unions left-to-right. The overall estimate of $\beta_{F_{ROH}}$ is shown as a solid black line. Subjects with kinship equal to offspring of full-sibling or parent–child unions are significantly shorter than those of avuncular or half-sibling unions who in turn are significantly shorter than those of first-cousin unions. **b** Linear decrease in odds of ever having children with increasing $F_{ROH}$. Linear model approximations of ln(Odds-Ratio) for ever having children (1 = parous, 0 = childless) are plotted in bins of increasing $F_{ROH}$. A strong relationship is evident, extending beyond the offspring of first cousins. **c** ROH effects are consistent in adoptees. The ratios of effect estimates, $\beta_{F_{ROH}}$, between adoptees and all individuals are presented by trait. All traits are directionally consistent and overall show a strongly significant difference from zero (average = 0.78, 95% CI 0.56–1.00, $p = 2 \times 10^{-12}$). FEV1 forced expiratory volume in one second. **d** ROH effects are consistent in full siblings. The ratios of effect estimates within full siblings to effects in all individuals ($\beta_{F_{ROH\_wSibs}} : \beta_{F_{ROH}}$) are presented by trait. Twenty-three of 29 estimates are directionally consistent and overall show a significant difference from zero (average = 0.78, 95% CI 0.53–1.04, $p = 7 \times 10^{-10}$). BMI body mass index. All errors bars represent 95% confidence intervals

significance. A meta-analysis of the ratio $\widehat{\beta}_{F_{\mathrm{ROH\_wSibs}}} : \widehat{\beta}_{F_{\mathrm{ROH}}}$ for all traits is significantly greater than zero (Fig. 5d; average = 0.78, 95% CI 0.53–1.04, $p = 7 \times 10^{-10}$), indicating a substantial fraction of these effects is genetic in origin. However, for both adoptees and siblings, the point estimates are less than one, suggesting that non-genetic factors probably contribute a small, but significant, fraction of the observed effects.

## Discussion

Our results reveal inbreeding depression to be broad in scope, influencing both complex traits related to evolutionary fitness and others where the pattern of selection is less clear. While studies of couples show optimal fertility for those with distant kinship[25,26], fewer have examined reproductive success as a function of individual inbreeding. Those that did are orders of magnitude smaller in size than the present study, suffer the attendant drawbacks of pedigree analysis, and have found mixed results[27–29]. Our genomic approach also reveals that in addition to socio-demographic factors and individual choice, recessive genetic effects have a significant influence on whether individuals reproduce. The discordant effects on fertility and education demonstrate that this is not just a result of genetic correlations between the two domains[30].

The effects we see on fertility might be partially mediated through a hitherto unknown effect of autozygosity on decreasing the prevalence of risk-taking behaviours. Significant effects of autozygosity are observed for self-reported risk taking, speeding on motorways, alcohol and smoking behaviour, age at first sexual intercourse and number of sexual partners. Independent evidence for a shared genetic architecture between risk-taking and fertility traits comes from analysis of genetic correlations using LD-score regression in UKB (Supplementary Table 1). The core fertility traits, ever had children and number of children, are strongly genetically correlated ($r_{\mathrm{G}} = 0.93$; $p < 10^{-100}$). Genetic correlations with ever-smoking and self-reported risk-taking are lower, but also significant: 0.23–0.27, $p < 10^{-10}$. Age at first sex is strongly genetically correlated both with the fertility traits, ($r_{\mathrm{G}} = 0.53$–0.57), and number of sexual partners, ever-smoking and risk-taking[30] ($r_{\mathrm{G}} = 0.42$–0.60).

Reproductive traits are understandable targets of natural selection, as might be walking speed, grip strength, overall health, and visual and auditory acuity. While we cannot completely exclude reverse causality, whereby a less risk-taking, more conservative, personality is associated with greater likelihood of consanguineous marriage, we note that the effects are consistent for ROH < 5 Mb, which are less confounded with mate choice, due to their more distant pedigree origins (Supplementary Fig. 11a). This group of traits also shows similar evidence for unconfounded effects in the analysis of adoptees and full siblings (Fig. 5c, d; Supplementary Data 16) and the signals remained after correcting for religious activity or education.

On the other hand, for some traits that we expected to be influenced by ROH, we observed no effect. For example, birth weight is considered a key component of evolutionary fitness in mammals, and is influenced by genomic homozygosity in deer[31]; however, no material effect is apparent here (Supplementary Data 10). Furthermore, in one case, ROH appear to provide a beneficial effect: increasing $F_{\mathrm{ROH}}$ significantly decreases total and LDL-cholesterol in men, and may thus be cardio-protective in this regard.

Our multivariate models show that homozygosity at common SNPs outside of ROH has little influence on traits, and that the effect rather comes from ROH over 1.5 Mb in length. This suggests that genetic variants causing inbreeding depression are almost entirely rare, consistent with the dominance hypothesis[1].

The alternative hypothesis of overdominance, whereby positive selection on heterozygotes has brought alleles to intermediate frequencies, would predict that more common homozygous SNPs outside long ROH would also confer an effect. The differential provides evidence in humans that rare recessive mutations underlie the quantitative effects of inbreeding depression.

Previous studies have shown that associations observed between $F_{\mathrm{ROH}}$ and traits do not prove a causal relationship[14,24]. Traditional Genome-wide Association Studies (GWAS) can infer causality because, in the absence of population structure, genetic variants (SNPs) are randomly distributed between, and within, different social groups. However, this assumption does not hold in studies of inbreeding depression, where, even within a genetically homogeneous population, social groups may have differing attitudes towards consanguinity, and therefore different average $F_{\mathrm{ROH}}$ and, potentially, different average trait values. We therefore present a number of analyses that discount social confounding as a major factor in our results. Firstly, we show that the effects are consistent across diverse populations, including those where ROH burden is driven by founder effects rather than cultural practices regarding marriage. Effects are also consistent across a 20-fold range of $F_{\mathrm{ROH}}$: from low levels, likely unknown to the subject, to extremely high levels only seen in the offspring of first-degree relatives. Secondly, we show that the effects of ROH are consistent in direction and magnitude among adopted individuals, and also for short ROH which are not informative about parental relatedness. Finally, we introduce a within-siblings method, independent of all confounders, that confirms a genetic explanation for most of the observed effects. Variation in $F_{\mathrm{ROH}}$ between siblings is caused entirely by random Mendelian segregation; we show that higher $F_{\mathrm{ROH}}$ siblings experience poorer overall health and lower reproductive success, as well as other changes consistent with population-wide estimates. Nevertheless, average effect sizes from both adoptees and siblings are 20% smaller than population-wide estimates, confirming the importance of accounting for social confounding in future studies of human inbreeding depression.

Our results reveal five large groups of phenotypes sensitive to inbreeding depression, including some known to be closely linked to evolutionary fitness, but also others where the connection is, with current knowledge, more surprising. The effects are mediated by ROH rather than homozygosity of common SNPs, causally implicating rare recessive variants rather than overdominance as the most important underlying mechanism. Identification of these recessive variants will be challenging, but analysis of regional ROH and in particular using whole-genome sequences in large cohorts with sufficient variance in autozygosity will be the first step. Founder populations or those which prefer consanguineous marriage will provide the most power to understand this fundamental phenomenon.

see Supplementary Data.

## Methods

**Overview.** Our initial aim was to estimate the effect of $F_{\mathrm{ROH}}$ on 45 quantitative traits and to assess whether any of these effects differed significantly from zero. Previous work[7,11] has shown that inbreeding coefficients are low in most human populations, and that very large samples are required to reliably estimate the genetic effects of inbreeding[13]. To maximise sample size, a collaborative consortium (ROHgen[6]) was established, and research groups administering cohorts with SNP chip genotyping were invited to participate. To ensure that all participants performed uniform and repeatable analyses, a semi-automated software pipeline was developed and executed locally by each research group. This software pipeline required cohorts to provide only quality-controlled genotypes (in plink binary format) and standardised phenotypes (in plain-text) and used standard software (R, PLINK[12,32], KING[33]) to perform the analyses described below. Results from each cohort were returned to the central ROHgen analysts for meta-analysis.

During the initial meta-analysis, genotypes were released for >500,000 samples from the richly phenotyped UK Biobank (UKB)[10]. It was therefore decided to add a

further 34 quantitative phenotypes and 21 binary traits to the ROHgen analysis. Many of these additional traits were unique to UKB, although 7 were also available in a subset of ROHgen cohorts willing to run additional analyses. In total, the effect of $F_{ROH}$ was tested on 100 traits and therefore experiment-wise significance was defined as $5 \times 10^{-4}$ (=0.05/100).

**Cohort recruitment.** In total, 119 independent, genetic epidemiological study cohorts were contributed to ROHgen. Of these, 118 were studies of adults and contributed multiple phenotypes, while 1 was a study of children and contributed only birth weight. To minimise any potential confounding or bias caused by within-study heterogeneity, studies were split into single-ethnicity sub-cohorts wherever applicable. Each sub-cohort was required to use only one genotyping array and be of uniform ancestry and case-status. For example, if a study contained multiple distinct ethnicities, sub-cohorts of each ancestry were created and analysed separately. At minimum, ancestry was defined on a sub-continental scale (i.e. European, African, East Asian, South Asian, West Asian, Japanese, and Hispanic were always analysed separately) but more precise separation was used when deemed necessary, for example, in cohorts with large representation of Ashkenazi Jews. In case-control studies of disease, separate sub-cohorts were created for cases and controls and phenotypes associated with disease status were not analysed in the case cohort: for example, fasting plasma glucose was not analysed in Type 2 diabetes case cohorts. Occasionally, cohorts had been genotyped on different SNP genotyping microarrays and these were also separated into sub-cohorts. There was one exception (deCODE) to the single microarray rule, where the intersection between all arrays used exceeded 150,000 SNPs. In this cohort the genotype data from all arrays was merged since the correspondence between $F_{ROH}$ for the individual arrays and $F_{ROH}$ the intersection dataset was found to be very strong ($\beta_{merged,hap} = 0.98$, $r^2 = 0.98$; $\beta_{merged,omni} = 0.97$, $r^2 = 0.97$). Dividing studies using these criteria yielded 234 sub-cohorts. Details of phenotypes contributed by each cohort are available in Supplementary Data 4.

**Ethical approval.** Data from 119 independent genetic epidemiology studies were included. All subjects gave written informed consent for broad-ranging health and genetic research and all studies were approved by the relevant research ethics committees or boards. PubMed references are given for each study in Supplementary Data 2.

**Genotyping.** All samples were genotyped on high-density (minimum 250,000 markers), genome-wide SNP microarrays supplied by Illumina or Affymetrix. Genotyping arrays with highly variable genomic coverage (such as Exome chip, Metabochip, or Immunochip) were judged unsuitable for the ROH calling algorithm and were not permitted. Imputed genotypes were also not permitted; only called genotypes in PLINK binary format were accepted. Each study applied their own GWAS quality controls before additional checks were made in the common analysis pipeline: SNPs with >3% missingness or MAF <5% were removed, as were individuals with >3% missing data. Only autosomal genotypes were used for the analyses reported here. Additional, cohort-specific, genotyping information is available in Supplementary Data 2.

**Phenotyping.** In total, results are reported for 79 quantitative traits and 21 binary traits. These traits were chosen to represent different domains of health and reproductive success, with consideration given to presumed data availability. Many of these traits have been the subject of existing genome-wide association meta-analyses (GWAMA), and phenotype modelling, such as inclusion of relevant covariates, was copied from the relevant consortia (GIANT for anthropometry, EGG for birth weight, ICBP for blood pressures, MAGIC for glycaemic traits, CHARGE-Cognitive, -Inflammation and -Haemostasis working groups for cognitive function, CRP, fibrinogen, CHARGE-CKDgen for eGFR, CHARGE-ReproGen for ages at menarche and menopause, Blood Cell & HaemGen for haematology, GUGC for urate, RRgen, PRIMA, QRS & QT-IGC for electrocardiography, GLGC for classical lipids, CREAM for spherical equivalent refraction, Spirometa for lung function traits, and SSGAC for educational attainment and number of children ever born). Further information about individual phenotype modelling is available in Supplementary Note 1 and Supplementary Data 8.

**ROH calling.** Runs of homozygosity (ROH) of >1.5 Mb in length were identified using published methods[6,11]. In summary, SNPs with minor allele frequencies below 5% were removed, before continuous ROH SNPs were identified using PLINK with the following parameters: homozyg-window-snp 50; homozyg-snp 50; homozyg-kb 1500; homozyg-gap 1000; homozyg-density 50; homozyg-window-missing 5; homozyg-window-het 1. No linkage disequilibrium pruning was performed. These parameters have been previously shown to call ROH that correspond to autozygous segments in which all SNPs (including those not present on the chip) are homozygous-by-descent, not chance arrangements of independent homozygous SNPs, and inbreeding coefficient estimates calculated by this method ($F_{ROH}$) correlate well with pedigree-based estimates ($F_{PED}$)[11]. Moreover, they have also been shown to be robust to array choice[6].

**Calculating estimators of $F$.** For each sample, two estimates of the inbreeding coefficient ($F$) were calculated, $F_{ROH}$ and $F_{SNP}$. We also calculated three additional measures of homozygosity: $F_{ROH<5Mb}$, $F_{ROH>5Mb}$ and $F_{SNP\_outsideROH}$.

$F_{ROH}$ is the fraction of each genome in ROH >1.5 Mb. For example, in a sample for which PLINK had identified $n$ ROH of length $l_i$ (in Mb), $i \in \{1..n\}$, then $F_{ROH}$ was then calculated as

$$F_{ROH} = \frac{\sum_{i=1}^{n} l_i}{3Gb},$$ (1)

where $F_{ROH<5Mb}$ and $F_{ROH>5Mb}$ are the genomic fractions in ROH of length >5 Mb, and in ROH of length <5 Mb (but >1.5 Mb), respectively, and the length of the autosomal genome is estimated at 3 gigabases (Gb). It follows from this definition that

$$F_{ROH} = F_{ROH>5Mb} + F_{ROH<5Mb}.$$ (2)

Single-point inbreeding coefficients can also be estimated from individual SNP homozygosity without any reference to a genetic map. For comparison with $F_{ROH}$, a method of moments estimate of inbreeding coefficient was calculated[34], referred to here as $F_{SNP}$, and implemented in PLINK by the command–het.

$$F_{SNP} = \frac{O(HOM) - E(HOM)}{N - E(HOM)},$$ (3)

where $O(HOM)$ is the observed number of homozygous SNPs, $E(HOM)$ is the expected number of homozygous SNPs, i.e. $\sum_{i=1}^{N} (1 - 2p_i q_i)$, and $N$ is the total number of non-missing genotyped SNPs.

$F_{ROH}$ and $F_{SNP}$ are strongly correlated, especially in cohorts with significant inbreeding, since both are estimates of $F$. To clarify the conditional effects of $F_{ROH}$ and $F_{SNP}$, an additional measure of homozygosity, $F_{SNPoutsideROH}$, was calculated to describe the SNP homozygosity observed outside ROH.

$$F_{SNP_{outsideROH}} = \frac{O'(HOM) - E'(HOM)}{N' - E'(HOM)},$$ (4)

where

$$O'(HOM) = O(HOM) - N_{SNP\_ROH},$$ (5)

$$E'(HOM) = \left(\frac{N - N_{ROH}}{N}\right) * E(HOM),$$ (6)

$$N' = N - N_{ROH}$$ (7)

And $N_{SNP\_ROH}$ is the number of homozygous SNPs found in ROH. Note that:

$$F_{SNPoutsideROH} \approx F_{SNP} - F_{ROH}$$ (8)

A further single point estimator of the inbreeding coefficient, described by Yang et al.[20] as $\hat{F}^{III}$, is implemented in PLINK by the parameter –ibc (Fhat3) and was also calculated for all samples.

$$F_{GRM} = \hat{F}^{III} = \frac{1}{N} \sum_{i=1}^{N} \frac{(x_i^2 - (1 + 2p_i)x_i + 2p_i^2)}{2p_i(1 - p_i)},$$ (9)

where $N$ is the number of SNPs, $p_i$ is the reference allele frequency of the $i$th SNP in the sample population and $x_i$ is the number of copies of the reference allele.

**Effect size estimates for quantitative traits.** In each cohort of $n$ samples, for each of the quantitative traits measured in that cohort, trait values were modelled by

$$y = \beta_{F_{ROH}} * F_{ROH} + Xb + \varepsilon,$$ (10)

where $y$ is a vector ($n \times 1$) of measured trait values, $\beta_{F_{ROH}}$ is the unknown scalar effect of $F_{ROH}$ on the trait, $F_{ROH}$ is a known vector ($n \times 1$) of individual $F_{ROH}$, $b$ is a vector ($m \times 1$) of unknown fixed covariate effects (including a mean, $\mu$), $X$ in a known design matrix ($n \times m$) for the fixed effects, and $\varepsilon$ is an unknown vector ($n \times 1$) of residuals.

The $m$ fixed covariates included in each model were chosen with reference to the leading GWAMA consortium for that trait and are detailed in Supplementary Data 3. For all traits, these covariates included: age (and/or year of birth), sex, and at least the first 10 principal components of the genomic relatedness matrix (GRM). Where necessary, additional adjustments were made for study site, medications, and other relevant covariates (Supplementary Data 3).

For reasons of computational efficiency, it was decided to solve Eq. (10) in two steps. In the first step, the trait ($y$) was regressed on all fixed covariates to obtain the maximum likelihood solution of the model:

$$y = Xb + \varepsilon'.$$ (11)

All subsequent analyses were performed using the vector of trait residuals $\varepsilon'$, which may be considered as the trait values corrected for all known covariates.

In cohorts with a high degree of relatedness, mixed-modelling was used to correct for family structure, although, because ROH are not narrow-sense heritable, this was considered less essential than in Genome-Wide Association Studies. Equation (11) becomes

$$y = Xb + u + \varepsilon',$$ (12)

where $\mathbf{u}$ is an unknown vector ($n \times 1$) of polygenic effects with multivariate normal distribution of mean 0 and covariance matrix $\sigma_g^2 \mathbf{A}$, where $\mathbf{A}$ is the genomic relationship matrix (GRM). In these related cohorts, a GRM was calculated using PLINK v1.9 and Grammar+ residuals of Eq. (12) were estimated using GenABEL[35]. These Grammar+ residuals ($\mathbf{\varepsilon}'$) were used in subsequent analyses.

To estimate $\beta_{F_{ROH}}$ for each trait, trait residuals were regressed on $F_{ROH}$ to obtain the maximum likelihood (ML) solution of the model

$$\mathbf{\varepsilon}' = \mu + \beta_{F_{ROH}} * \mathbf{F_{ROH}} + \mathbf{\varepsilon}. \tag{13a}$$

The sex-specific estimates of $\beta_{F_{ROH}}$ (Supplementary Data 12) were obtained from Eq. (13) applied to the relevant sex.

For all traits, a corresponding estimates of $\beta_{F_{SNP}}$ and $\beta_{F_{GRM}}$ were obtained from the models

$$\mathbf{\varepsilon}' = \mu + \beta_{F_{SNP}} * \mathbf{F_{SNP}} + \mathbf{\varepsilon}, \tag{13b}$$

$$\mathbf{\varepsilon}' = \mu + \beta_{F_{GRM}} * \mathbf{F_{GRM}} + \mathbf{\varepsilon} \tag{14}$$

and the effects of different ROH lengths and of SNP homozygosity (Fig. 4b) were obtained from the model

$$\mathbf{\varepsilon}' = \mu + \left( \beta_1 * \mathbf{F_{SNP_{outsideROH}}} \right) + \left( \beta_2 * \mathbf{F_{ROH<5Mb}} \right) \\ + \left( \beta_3 * \mathbf{F_{ROH>5Mb}} \right) + \mathbf{\varepsilon}. \tag{15}$$

**Effect size estimates for binary traits.** Binary traits were analysed by two methods. The primary estimates of $\beta_{F_{ROH}}$ (Fig. 3b and Supplementary Data 10) were obtained from full logistic models:

$$g(E[\mathbf{y}]) = \mathbf{Xb}, \tag{16}$$

where $g()$ is the link function (logit), and where $F_{ROH}$ and all applicable covariates (Supplementary Datas 3, 8) were fitted simultaneously. Mixed modelling for family structure was not attempted in the logistic models since an accepted method was not apparent.

For all subsequent results, $\mathbf{y}$ was scaled by $1/\sigma_y^2$ and analysed by linear models, as for quantitative traits, including mixed-modelling where appropriate for family studies. This method of estimating binary traits with simple linear models gives asymptotically unbiased estimates of $\beta_{F_{ROH}}$ and se($\beta_{F_{ROH}}$) on the ln(Odds-Ratio) scale[36]. For all significant binary traits, a comparison of $\widehat{\beta}_{F_{ROH}}$ from the full model with $\widehat{\beta}_{F_{ROH}}$ from the linear model approximation is presented in Supplementary Fig. 8.

To give $\widehat{\beta}_{F_{ROH}}$ a more tangible interpretation, effect estimates are frequently quoted in the text as $\beta_{0.0625}$, i.e. the estimated effect in the offspring of first cousins, where 6.25% of the genome is expected to be autozygous.

**Religiosity and educational attainment as additional covariates.** To assess the importance of potential social confounders, proxy measures of socio-economic status and religiosity were separately included in Eq. (13) as additional covariates. The modified effect estimates ($\widehat{\beta}'_{F_{ROH}}$) were tested for significance (Supplementary Data 20) and compared to the uncorrected estimates ($\beta_{F_{ROH}}$) (Supplementary Fig. 10a, b).

Since Educational Attainment (EA) was measured in many cohorts, this was chosen as the most suitable proxy for socio-economic status. However, since $F_{ROH}$ is known to affect EA directly[6] any change in $\beta_{F_{ROH}}$ when conditioning on EA cannot be assumed to be entirely due to environmental confounding.

The analysis of religiosity was only carried out in UKB, where a rough proxy was available. Although no direct questions about religious beliefs were included, participants were asked about their leisure activities. In response to the question *Which of the following do you attend once a week or more often? (You can select more than one)*, 15.6% of UKB participants selected *Religious Group* from one of the seven options offered. In the models described, religiosity was coded as 1 for those who selected *Religious Group* and 0 for those who did not. Although this is likely to be an imperfect measure of actual religious belief it is currently the best available in a large dataset.

**Assortative mating.** Humans are known to mate assortatively for a number of traits including height and cognition[37], and so we sought to investigate if this could influence our results, for example, by the trait extremes being more genetically similar and thus the offspring more homozygous. We see no evidence for an effect of assortative mating on autozygosity, however. Firstly, a polygenic risk score for height (see Supplementary Note 1), which explains 18.7% of the phenotypic variance in height, was not associated with $F_{ROH}$ ($p = 0.77$; Supplementary Fig. 5). Secondly, linear relationships between traits and autozygosity extend out to very high $F_{ROH}$ individuals (Supplementary Figs. 9a–f). Samples in the highest $F_{ROH}$ group are offspring of genetically similar parents, very likely first or second degree relatives and, for example, the height of these samples is on average 3.4 cm [95% CI 2.5–4.3] shorter than the population mean. Assortative mating would suggest this

height deficit has been inherited from genetically shorter parents, but this would require an implausibly strong relationship between short stature and a propensity to marry a very close relative. Thirdly, the sex-specific effects we observe could only be explained by assortative mating if the additive heritability of these traits also differed by gender.

**Average trait values in groups of similar $F_{ROH}$.** In each cohort individuals were allocated to one of ten groups of similar $F_{ROH}$. The bounds of these groups were the same for all cohorts, specifically {0, 0.002, 0.0041, 0.0067, 0.0108, 0.0186, 0.0333, 0.06, 0.10, 0.18 and 1.0}. Within each group the mean trait residual ($\mathbf{\varepsilon}'$) and mean $F_{ROH}$ were calculated, along with their associated standard errors. Within each cohort the expectation of $\mathbf{\varepsilon}'$ is zero at the mean $F_{ROH}$, however as mean $F_{ROH}$ varies between cohorts (Fig. 2, Supplementary Data 5) it was necessary to express $\mathbf{\varepsilon}'$ relative to a common $F_{ROH}$ before meta-analysis. Hence, for this analysis only, the trait residuals ($\mathbf{\varepsilon}'$) were expressed relative to the $F_{ROH} = 0$ intercept, i.e. by subtracting $\mu$ from Eq. (13).

**Effect of $F_{ROH}$ within adoptees.** We compared $\beta_{F_{ROH\_ADOPTEE}}$ to cross-cohort $\beta_{F_{ROH}}$, not that from UKB alone, as we consider the latter to be a noisy estimate of the former; estimates in UKB are consistent with those from meta-analysis.

**Effect of $F_{ROH}$ within full-sibling families.** In a subset of cohorts, with substantial numbers of related individuals, further analyses were performed to investigate the effect of $F_{ROH}$ within full-sibling families. In each of these cohorts, all second-degree, or closer, relatives were identified using KING (parameters:–related–degree 2). Full-siblings were then selected as relative pairs with genomic kinship >0.175 and IBS0 >0.001. This definition includes monozygotic twins, who were intentionally considered as part of full-sibling families. Although monozygotic twins are expected to have identical $F_{ROH}$, they may not have identical trait values, and including additional trait measurements decreases the sampling error of the within-family variance estimate, hence increasing statistical power. Dizygotic twins were also included.

For each individual ($j$) with identified siblings, the values of $F_{ROH}$ and trait residual ($\mathbf{\varepsilon}'$) were calculated relative to their family mean (and called $F_j^{ROH\_wSibs}$ and $\varepsilon_j^{wSibs}$, respectively), i.e. for individual $j$ with $n$ full-siblings $S_k$ where $k \in \{1..n\}$

$$F_j^{ROHwSibs} = F_j^{ROH} - \frac{1}{(n+1)} \sum_{i \in \{j, S_k\}} F_i^{ROH}, \tag{17}$$

$$\varepsilon_j^{wSibs} = \varepsilon_j' - \frac{1}{(n+1)} \sum_{i \in \{j, S_k\}} \varepsilon'_i. \tag{18}$$

The effect of $F_{ROH}$ within-full-siblings ($\beta_{F_{ROH\_wSibes}}$) was estimated by linear regression of $\mathbf{\varepsilon}^{wSibs}$ on $\mathbf{F^{ROH\_wSibs}}$.

Importantly, the variation of $F_{ROH}$ within full-siblings is entirely caused by differences in Mendelian segregation, and is therefore completely independent of all possible confounders. Hence, the effect estimates obtained by this method are estimates of the genetic effects of $F_{ROH}$, unbiased by any possible confounder. Since confounding by social factors is a major concern in this field, methods that can definitively exclude this possibility are of critical importance.

**Between-cohort meta-analysis.** As is typical in genome-wide association meta-analyses (GWAMA), genetic effects were estimated within single-ethnicity sub-cohorts, and meta-analysis of the within-cohort effect sizes was used to combine results[38]. This established method eliminates any potential confounding caused by between-cohort associations between $F_{ROH}$ and traits.

Each cohort returned estimates and standard errors of: $\beta_{F_{ROH}}$, $\beta_{F_{SNP}}, \beta_{F_{ROH>Mb}}, \beta_{F_{ROH<Mb}}, \beta_{F\_outsideROH}, \beta_{F_{ROH\_wSibs}}$, as well as trait means ($\overline{\varepsilon'}$) and standard errors within each of 10 $F_{ROH}$ bins. The between-cohort mean of each of these 16 estimates was then determined by fixed-effect, inverse-variance meta-analysis using the R package metafor[39]. Results shown in Figs. 3–5 are meta-analysed averages of the within-cohort effects.

The meta-analysis was also run for various subsets of cohorts, stratified by ancestry as defined in Supplementary Data 18. Meta-analysis estimates from these groupings are shown in Supplementary Fig. 1.

**Median and 95% CI of a ratio.** In the analyses of adoptees (Fig. 5c), siblings (Fig. 5d) and potential confounders (Supplementary Figs. 10a, b) we wished to compare the effect estimates ($\beta_{F_{ROH}}$) from two different methods across a wide range of traits. The units of $\beta_{F_{ROH}}$ differ by trait so, to allow comparison across all traits, the unitless ratio of effect size estimates was calculated (for example $\beta_{F_{ROH\_wSibs}} : \beta_{F_{ROH}}$). Figure 5c, d and Supplementary Figs. 10a, b show the medians and 95% CI of these ratios. These were determined empirically by bootstrap since, although formulae exist for the mean and standard error of a ratio[40], the assumption of normality is violated when $\beta_{F_{ROH}}/se(\beta_{F_{ROH}})$ is not large.

**Genetic correlations in UK Biobank.** Genetic correlations were calculated using LD-Score Regression[41], implemented in LDSC v1.0.0 (https://github.com/bulik/ldsc). Summary statistics were parsed using default parameters in the LDSC

'munge_sumstats.py' script, extracting only variants present in the HapMap 3 reference panel.

**Accuracy of $F_{ROH}$ measures of inbreeding effects**. A recent paper suggested that ROH may overestimate inbreeding effects by as much as 162%[42]; however, this could only be the case if $F_{ROH}$ underestimates excess homozygosity at the causal loci by at least 162%. We do not believe this to be the case since the maximum $F_{ROH}$ measured in many cohorts is around 0.25 (the expectation in the offspring off first-degree relatives), and the effect size estimates from these samples are consistent with the overall estimates (Fig. 5c, d and Supplementary Fig. 9a–f). We note that Yengo et al. applied the ROH calling parameters used here to imputed data. These parameters have been validated for called genotype data[6] but not, to our knowledge, for the higher SNP density and error rate of imputed data (see also Supplementary Note 4). The simple method for detecting ROH used here was well suited to our study, since it could be easily implemented on over one million samples, and most of the variation in $F_{ROH}$ is caused by easily-identified long ROH.[43–45]

**Reporting summary**. Further information on research design is available in the Nature Research Reporting Summary linked to this article.

## Data availability

The meta-analysed data which support these findings are available as Supplementary Data files. Cohort-level summary statistics underlying all figures and tables are available in a publicly accessible dataset (https://doi.org/10.6084/m9.figshare.9731087). In the majority of cases we do not have consent to share individual-level data, although for UK Biobank this is available on request from https://www.ukbiobank.ac.uk/.

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

## Acknowledgements

This paper is the work of the ROHgen consortium. We thank the Sigma T2D Consortium, whose members are detailed in Supplementary Note 3. We thank the UK Biobank Resource, approved under application 19655; we acknowledge funding from the UK Medical Research Council Human Genetics Unit and MRC Doctoral Training Programme in Precision Medicine. We also thank Neil Robertson, Wellcome Trust Centre for Human Genetics, Oxford, for use of his author details management software, Authorial. Finally, we thank all the participants, researchers and funders of ROHgen cohorts. Cohort-specific acknowledgements are in Supplementary Data 2; personal acknowledgements and disclosures are in Supplementary Note 2. We thank Rachel Edwards for administrative assistance.

## Author Contributions

Directed consortium: T.E., P.K.J. and J.F.W. Central analyst: D.W.C., Y.O., N. Pirastu and P.K.J. Cohort PI: C.A.A.-S., M.L.A., T.A., F.A., E.B., G.J.d.B., E.P.B., S. Chanock, Y.-D.I.C., Z.C., R.M.v.D., M.S.D., N.D., P.E., B.I.F., C.A. Haiman, D.A.v.H., I.H., M.A.I., P.J., T. Kessler, K.-T.K., D.P.V.d.K., W.-P.K., J. Kuusisto, C.L., D.A.L., L.L., T.A.M., Y. Murakami, K.K.O., L.O., O.P., N. Poulter, P.P.P., L.Q.-M., K.R., D.C.R., S.S.R., P.M.R., L.J.S., P.J.S., W.H.-H.S., A. Stanton, J.M.S., L. Straker, T.T.-L., J.H.V., V.V., Y.X.W., N.J.W., C.S. Y., J.-M.Y., F.W.A., S.J.L.B., D.M.B., D.A.B., L.H.v.d.B., S.I.B., D.B., C. Bouchard, M.J.C., J.C.C., G.R.C., C.-Y.C., M. Ciullo, M. Cornelis, D. Cusi, G.D.-S., I.D., C.M.v.D., D.E., J. Erdmann, J.G.E., E.E., M.K.E., B.F., M.F., A.F., Y.F., P. Gasparini, C. Gieger, C. Gonzalez, S.F.A.G., L.R.G., L.G., V.G., U.G., A. Hamsten, P.v.d.H., C.-K.H., H. Hochner, S.C.H., V.W.V.J., Å.J., J.B.J., J.W.J., J.J., J. Kaprio, S.L.K., F.K., M. Kumari, M. Laakso, M. Laudes, W.L., N.G.M., W. März, G. Matullo, M.I.M., T.R.M., A. Metspalu, B.F.M., K.L. M., G.W.M., D.M.-K., P.B.M., K.E.N., C.O., A.J.O., C. Palmer, G.G.P., E.P., C.E.P., L.P., M. Pirastu, D.J.P., D.P., B.M.P., C.R., J.I.R., I.R., D.K.S., R.S., H. Schunkert, A.R.S., N. Small, E.-S.T., N.J.T., D.T., T.T., P.V., D.R.W., T.-Y.W., J.W., A.B.Z., M. Perola, P.K.M., A.G.U., J.S.K., D.I.C., R.J.F.L., N.F., C.H., J.R.B.P., T.E., K. Stefansson, M. Kubo, J.F.W. Cohort analyst: D.W.C., Y.O., K.H.S.M., D.M., I.G., H.M., K.L., J.H.Z., P.D., R.R., C. Schurmann, X.G., F. Giulianini, W. Zhang, C.M.-G., R.K., Y.B., T.M.B., C. Baumbach, G.B., M.J.B., M. Brumat, J.-F.C., D.L.C., D.A.E., C.F., H.G., M. Germain, S.D.G., H.G.d.H., S.E.H., E.H., A.H.-C., C.I., I.E.J., Y.J., T. Kacprowski, T. Karlsson, M.E.K., S.A.L., R.L.-G., A. Mahajan, W. Meng, M.E.M., P.J.v.d.M., M.Munz, T.N., T.P., G. Prasad, R.B.P., T.D.S.P., F. Rizzi, E.S., B.R.S., D.S., L. Skotte, A.V.S., A.v.dS., C.N.S., R.J.S., S.M.T., S.T., C.T., N.V., C.V., L.W., H.R.W., R.E.W., L.R.Y., J. Yao, N.A.Y., W. Zhao, A.A.A., S.A., M. Akiyama, M. Alver, G. Chen, M. Cocca, M.P.C., G. Cugliari, F.R.D., G.E.D., G.G., A.G., M. Gögele, M. Graff, E.G.-H., A. Halevy, D.A.v.H., J.H., Y.K., M. Kanai, N.D.K., M. Loh, S.L., Y.L., J'a. L., N.M., X.W.M., M. Mezzavilla, A. Moore, H.M.-M., M.A.N., C.A.R., A.R., D.R., M.S.-L., W.R.S., B.S., J.v.s., S.S., S.R.S., T.T.S., A. Tillander, E.V., L. Zeng, N.A., L. Benjamin, L.F.B., J.P.B., J.A.B., S. Carmi, G.R.C., M. Cornelis, D. Cusi, R.D., D.E., J.D.F., M.F., S.F., P. Goyette, S.C.H., Å.J., S.W.v.d.L., J. Lahti, R.A.L., S.E.M., K.E.N., J.RO'c., E.P., L.P., J.D. R., F. Rivadeneira, C.R., R.A.S., X.S., J.A.S., C.A.W., J. Yang, L.Y., D.I.C., N.F., R.G.W., J.R. B.P., T.E., A. Helgason. Provided data: Y.O., C. Schurmann, W. Zhang, C.M.-G., Y.B., A.M.D., K.R.v.E., C.F., H.G., M.E.K., K. Matsuda, R.B.P., F. Rizzi, E.S., M.C.S., A.V.S., S.M. T., N.V., A.A.A., C.A.A.-S., M.L.A., M.A.A., A.R.B., E.B., J.B.B., G.J.d.B., E.P.B., L. Broer, H.C., S. Chanock, M.-L.C., G. Chen, Y.-D.I.C., Y.-F.C., J.C., M.S.D., K.D., M.D., A.P.D., N.D., S.S.E., J. Elliott, R.E., J.F.F., K.F., B.I.F., M. Gögele, M.O.G., S.G., D.F.G., K.G., B.G., Y.G., S.P.H., C.A. Haiman, T.B.H., M. Hedayati, M. Hirata, I.H., C.A. Hsiung, Y.-J.H., M.A.I., A.J., P.J., Y.K., C.C.K., W.-P.K., I.K., B.K.K., J. Kuusisto, L.J.L., D.A.L., I.-T.L., W.-J. L., M.M.L., J. Liu, S.J.L., R.M., A.W.M., P.M., G. Másson, C.M., T. Meitinger, L.M., I.Y.M., Y. Momozawa, T.A.M., A.C.M., T. Muka, A.D.M., R.d.M., J.C.M., M.A.N., M.N., M.J.N., I.M.N., L.O., S.P., G. Pálsson, J.S.P., C. Pattaro, A.P., O.P., N. Poulter, L.Q.-M., K.R., S.R., D.C.R., W.v.R., F.J.A.v.R., C. Sabanayagam, C.F.S., V.S., K. Sandow, B.S.-K., P.J.S., W.H.- H.S., Y. Shi, S.R.S., J.K.S., J.R.S., B.H.S., A. Stanton, L. Stefansdottir, L. Straker, P.S., G.S., M.A.S., A.M.T., K.D.T., N.T., Y.-C.T., G.T., U.T., R.P.T., T.T.-L., I.T., S.V., J.H.V., V.V., U.V., E.V., S.M.W., M.W., G.S.W., S.W., C.S.Y., J.-M.Y., L. Zhang, J.Z., S.J.L.B., D.M.B., S.I.B., A.C., M.J.C., D. Cesarini, J.C.C., G.R.C., M. Cornelis, D. Cusi, G.D.-S., I.D., R.D., D.E., J. Erdmann, J.G.E., E.E., M.K.E., B.F., M.F., Y.F., P. Gasparini, C. Gonzalez, S.F.A.G., L.G., V.G., C.-K.H., A.A.H., H. Hochner, H. Huikuri, S.C.H., V.W.V.J., P.L.D.J., M.J., J.W. J., J. Kaprio, S.L.K., M. Laakso, S.W.v.d.L., J.Lahti, W.L., N.G.M., G. Matullo, B.F.M., K.L. M., G.W.M., P.B.M., D.R.N., A.J.O., W.P., C. Palmer, C.E.P., L.P., P.A.P., T.J.P., D.J.P., D.P., B.M.P., J.D.R., F. Rivadeneira, C.R., J.I.R., I.R., D.K.S., N. Sattar, H. Schunkert, A. Teumer, N.J.T., T.T., D.R.W., J.B.W., C.W., J.W., A.B.Z., M. Perola, P.K.M., A.G.U., J.S.K., D.I.C., L.F., C.S.H., C.H., R.G.W., T.E. Contributed to manuscript: D.W.C., C. Schurmann, S.E.H., S.M.T., E.P.B., J.C., A.P.D., X.G., S.P.H., D.A.v.H., P.J., W.-P.K., D.A.L., S.J.L., A.P., J.M.S., A.M.T., J.-M.Y., J.Z., I.D., R.D., Y.F., J.W.J., C.R., J.I.R., M.Perola, R.J.F.L., P.K.J., J.F.W. Wrote manuscript: D.W.C. and J.F.W.

## Competing interests

M.L.A. is an employee of Genentech, a member of The Roche Group. D.A.L. has received support from several national and international government and charity funders, as well as Roche Diagnostics and Medtronic for work unrelated to this publication. M.I.M.: The views expressed in this article are those of the author(s) and not necessarily those of the NHS, the NIHR, or the Department of Health. He has served on advisory panels for Pfizer, NovoNordisk, Zoe Global; has received honoraria from Merck, Pfizer, Novo-Nordisk and Eli Lilly; has stock options in Zoe Global; has received research funding from Abbvie, Astra Zeneca, Boehringer Ingelheim, Eli Lilly, Janssen, Merck, Novo-Nordisk, Pfizer, Roche, Sanofi Aventis, Servier & Takeda. As of June 2019, M.Mc.C. is an employee of Genentech, and holds stock in Roche. T. Muka is now working as medical specialist at Novo Nordisk. O.P. is owner of Gen-info Ltd. Gen-info Ltd provided support in the form of salaries and financial gains for author O.P., but did not have any additional role in selection of the journal or preparation of this manuscript. N. Poulter received financial support from several pharmaceutical companies which manufacture either blood pressure lowering or lipid lowering agents, or both, and consultancy fees. V.S. has participated in a congress trip sponsored By Novo Nordisk. P.J.S. has received research awards from Pfizer Inc. M.J.C. is Chief Scientist for Genomics England, a UK government company. B.M.P. serves on the DSMB of a clinical trial funded by Zoll LifeCor and on the Steering Committee of the Yale Open Data Access Project funded by Johnson & Johnson. A.R.S. is an employee of Regeneron Pharmaceutical Inc. The remaining authors declare no competing interests.

## Additional information

David W Clark [1], Yukinori Okada [2,3,4], Kristjan H S Moore [5], Dan Mason [6], Nicola Pirastu [1], Ilaria Gandin [7,8], Hannele Mattsson [9,10], Catriona L K Barnes [1], Kuang Lin [11], Jing Hua Zhao [12,13], Patrick Deelen [14], Rebecca Rohde [15], Claudia Schurmann [16], Xiuqing Guo [17], Franco Giulianini [18], Weihua Zhang [19,20], Carolina Medina-Gomez [21,22,23], Robert Karlsson [24], Yanchun Bao [25], Traci M Bartz [26], Clemens Baumbach [27], Ginevra Biino [28], Matthew J Bixley [29], Marco Brumat [8], Jin-Fang Chai [30], Tanguy Corre [31,32,33], Diana L Cousminer [34,35], Annelot M Dekker [36], David A Eccles [37,38], Kristel R van Eijk [36], Christian Fuchsberger [39], He Gao [19,40], Marine Germain [41,42], Scott D Gordon [43], Hugoline G de Haan [44], Sarah E Harris [45,46], Edith Hofer [47,48], Alicia Huerta-Chagoya [49], Catherine Igartua [50], Iris E Jansen [51,52], Yucheng Jia [17,327], Tim Kacprowski [53,54], Torgny Karlsson [55], Marcus E Kleber [56], Shengchao Alfred Li [57], Ruifang Li-Gao [44], Anubha Mahajan [58], Koichi Matsuda [59], Karina Meidtner [60,61], Weihua Meng [62],

May E Montasser[63,64], Peter J van der Most[65], Matthias Munz[66,67,68,69], Teresa Nutile[70],
Teemu Palviainen[71], Gauri Prasad[72], Rashmi B Prasad[73], Tallapragada Divya Sri Priyanka[74],
Federica Rizzi[75,76], Erika Salvi[76,77], Bishwa R Sapkota[78], Daniel Shriner[79], Line Skotte[80], Melissa C Smart[25],
Albert Vernon Smith[81,82], Ashley van der Spek[22], Cassandra N Spracklen[83], Rona J Strawbridge[84,85],
Salman M Tajuddin[86], Stella Trompet[87,88], Constance Turman[89,90], Niek Verweij[91], Clara Viberti[92],
Lihua Wang[93], Helen R Warren[94,95], Robyn E Wootton[96,97], Lisa R Yanek[98], Jie Yao[17], Noha A Yousri[99,100],
Wei Zhao[101], Adebowale A Adeyemo[79], Saima Afaq[19], Carlos Alberto Aguilar-Salinas[102,103],
Masato Akiyama[3,104], Matthew L Albert[105,106,107,108], Matthew A Allison[109], Maris Alver[110],
Tin Aung[111,112,113], Fereidoun Azizi[114], Amy R Bentley[79], Heiner Boeing[115], Eric Boerwinkle[116], Judith B Borja[117],
Gert J de Borst[118], Erwin P Bottinger[16,119], Linda Broer[21], Harry Campbell[1], Stephen Chanock[120],
Miao-Li Chee[111], Guanjie Chen[79], Yii-Der I Chen[17], Zhengming Chen[11], Yen-Feng Chiu[121],
Massimiliano Cocca[122], Francis S Collins[123], Maria Pina Concas[122], Janie Corley[45,124],
Giovanni Cugliari[92], Rob M van Dam[30,125,126], Anna Damulina[47], Maryam S Daneshpour[127],
Felix R Day[12], Graciela E Delgado[56], Klodian Dhana[22,126,128], Alexander S F Doney[129], Marcus Dörr[130,131],
Ayo P Doumatey[79], Nduna Dzimiri[132], S Sunna Ebenesersdóttir[5,133], Joshua Elliott[19],
Paul Elliott[19,40,134,135,136], Ralf Ewert[130], Janine F Felix[22,23,137], Krista Fischer[110], Barry I Freedman[138],
Giorgia Girotto[8,139], Anuj Goel[58,140], Martin Gögele[39], Mark O Goodarzi[141], Mariaelisa Graff[15],
Einat Granot-Hershkovitz[142], Francine Grodstein[89], Simonetta Guarrera[92], Daniel F Gudbjartsson[5,143],
Kamran Guity[127], Bjarni Gunnarsson[5], Yu Guo[144], Saskia P Hagenaars[45,124,145], Christopher A Haiman[146],
Avner Halevy[142], Tamara B Harris[86], Mehdi Hedayati[127], David A van Heel[147], Makoto Hirata[148],
Imo Höfer[149], Chao Agnes Hsiung[121], Jinyan Huang[150], Yi-Jen Hung[151,152], M Arfan Ikram[22],
Anuradha Jagadeesan[5,133], Pekka Jousilahti[153], Yoichiro Kamatani[3,154], Masahiro Kanai[2,3,155],
Nicola D Kerrison[12], Thorsten Kessler[156], Kay-Tee Khaw[157], Chiea Chuen Khor[111,158],
Dominique P V de Kleijn[118], Woon-Puay Koh[30,159], Ivana Kolcic[160], Peter Kraft[126], Bernhard K Krämer[56],
Zoltán Kutalik[32,33], Johanna Kuusisto[161,162], Claudia Langenberg[12], Lenore J Launer[86],
Deborah A Lawlor[96,163,164], I-Te Lee[165,166,167], Wen-Jane Lee[168], Markus M Lerch[169], Liming Li[170],
Jianjun Liu[125,158], Marie Loh[19,171,172], Stephanie J London[173], Stephanie Loomis[174], Yingchang Lu[16],
Jian'an Luan[12], Reedik Mägi[110], Ani W Manichaikul[175], Paolo Manunta[176], Gísli Másson[5], Nana Matoba[3],
Xue W Mei[11], Christa Meisinger[177], Thomas Meitinger[178,179,180], Massimo Mezzavilla[139], Lili Milani[181],
Iona Y Millwood[11], Yukihide Momozawa[182], Amy Moore[120], Pierre-Emmanuel Morange[183,184],
Hortensia Moreno-Macías[185], Trevor A Mori[186], Alanna C Morrison[187], Taulant Muka[22,188],
Yoshinori Murakami[189], Alison D Murray[190], Renée de Mutsert[44], Josyf C Mychaleckyj[175],
Mike A Nalls[191,192], Matthias Nauck[131,193], Matt J Neville[194,195], Ilja M Nolte[65], Ken K Ong[12,196],
Lorena Orozco[197], Sandosh Padmanabhan[198], Gunnar Pálsson[5], James S Pankow[199], Cristian Pattaro[39],
Alison Pattie[124], Ozren Polasek[160,200], Neil Poulter[201,202], Peter P Pramstaller[39],
Lluis Quintana-Murci[203,204,205], Katri Räikkönen[206], Sarju Ralhan[207], Dabeeru C Rao[208],
Wouter van Rheenen[36], Stephen S Rich[175], Paul M Ridker[18,209], Cornelius A Rietveld[210,211],
Antonietta Robino[122], Frank J A van Rooij[22], Daniela Ruggiero[70,212], Yasaman Saba[213],
Charumathi Sabanayagam[111,112,113], Maria Sabater-Lleal[85,214], Cinzia Felicita Sala[215],
Veikko Salomaa[216], Kevin Sandow[17], Helena Schmidt[213], Laura J Scott[217], William R Scott[19],
Bahareh Sedaghati-Khayat[127], Bengt Sennblad[85,218], Jessica van Setten[219], Peter J Sever[201],
Wayne H-H Sheu[152,165,220,221], Yuan Shi[111], Smeeta Shrestha[74,222], Sharvari Rahul Shukla[223,224],
Jon K Sigurdsson[5], Timo Tonis Sikka[110], Jai Rup Singh[225], Blair H Smith[226], Alena Stančáková[161],

Alice Stanton [227], John M Starr[45,228,327], Lilja Stefansdottir[5], Leon Straker [229], Patrick Sulem [5], Gardar Sveinbjornsson[5], Morris A Swertz [14], Adele M Taylor[124], Kent D Taylor [17], Natalie Terzikhan[22,230], Yih-Chung Tham[111,112], Gudmar Thorleifsson[5], Unnur Thorsteinsdottir[5,82], Annika Tillander[24], Russell P Tracy[231], Teresa Tusié-Luna[232,233], Ioanna Tzoulaki [19,40,234], Simona Vaccargiu[235], Jagadish Vangipurapu[161], Jan H Veldink [36], Veronique Vitart [236], Uwe Völker [53,131], Eero Vuoksimaa[237], Salma M Wakil[132], Melanie Waldenberger[27], Gurpreet S Wander [238], Ya Xing Wang [239], Nicholas J Wareham[12], Sarah Wild [240], Chittaranjan S Yajnik[241], Jian-Min Yuan [242], Lingyao Zeng[156], Liang Zhang[111], Jie Zhou[79], Najaf Amin[22], Folkert W Asselbergs [243,244,245,246], Stephan J L Bakker[247], Diane M Becker[98], Benjamin Lehne[19], David A Bennett[248,249], Leonard H van den Berg[36], Sonja I Berndt[120], Dwaipayan Bharadwaj[250], Lawrence F Bielak[101], Murielle Bochud[32], Mike Boehnke [217], Claude Bouchard [251], Jonathan P Bradfield[252,253], Jennifer A Brody [254], Archie Campbell [46], Shai Carmi[142], Mark J Caulfield [94,95], David Cesarini[255,256], John C Chambers[19,20,40,257,258], Giriraj Ratan Chandak[74], Ching-Yu Cheng[111,112,113], Marina Ciullo[70,212], Marilyn Cornelis[259], Daniele Cusi[76,260,261], George Davey Smith [96,164], Ian J Deary[45,124], Rajkumar Dorajoo [158], Cornelia M van Duijn[11,22], David Ellinghaus [262], Jeanette Erdmann [66], Johan G Eriksson[263,264,265,266,267], Evangelos Evangelou [19,234], Michele K Evans[86], Jessica D Faul[268], Bjarke Feenstra [80], Mary Feitosa [93], Sylvain Foisy[269], Andre Franke [262], Yechiel Friedlander[142], Paolo Gasparini[8,139], Christian Gieger[27,61], Clicerio Gonzalez[270], Philippe Goyette[269], Struan F A Grant [35,252,271], Lyn R Griffiths [37], Leif Groop [71,73], Vilmundur Gudnason [81,82], Ulf Gyllensten[55], Hakon Hakonarson[252,271], Anders Hamsten[272], Pim van der Harst [91], Chew-Kiat Heng [273,274], Andrew A Hicks [39], Hagit Hochner[142], Heikki Huikuri[275], Steven C Hunt[99,276], Vincent W V Jaddoe [22,23,137], Philip L De Jager [277,278], Magnus Johannesson [279], Åsa Johansson [55], Jost B Jonas [239,280], J Wouter Jukema [87], Juhani Junttila[275], Jaakko Kaprio [71,281], Sharon L.R. Kardia[101], Fredrik Karpe[194,282], Meena Kumari[25], Markku Laakso [161,162], Sander W van der Laan [283], Jari Lahti [206,284], Matthias Laudes[285], Rodney A Lea[37], Wolfgang Lieb[286], Thomas Lumley[287], Nicholas G Martin [43], Winfried März[56,288,289], Giuseppe Matullo[92], Mark I McCarthy[58,194,282], Sarah E Medland [43], Tony R Merriman [29], Andres Metspalu [110], Brian F Meyer[290], Karen L Mohlke [83], Grant W Montgomery [43,291], Dennis Mook-Kanamori[44,292], Patricia B Munroe [94,95], Kari E North[15], Dale R Nyholt [43,293], Jeffery R O'connell[63,64], Carole Ober [50], Albertine J Oldehinkel [294], Walter Palmas[295], Colin Palmer [296], Gerard G Pasterkamp[149], Etienne Patin[203,204,205], Craig E Pennell[297,298], Louis Perusse [299,300], Patricia A Peyser[101], Mario Pirastu[301], Tinca J.C. Polderman [51], David J Porteous [45,46], Danielle Posthuma[51,302], Bruce M Psaty[303,304], John D Rioux [269,305], Fernando Rivadeneira [21,22,23], Charles Rotimi [79], Jerome I Rotter [17], Igor Rudan [1], Hester M Den Ruijter[306], Dharambir K Sanghera[78,307], Naveed Sattar [198], Reinhold Schmidt[47], Matthias B Schulze[60,61,308], Heribert Schunkert[156,309], Robert A Scott[12], Alan R Shuldiner[63,64,310], Xueling Sim [30], Neil Small[311], Jennifer A Smith [101,268], Nona Sotoodehnia[312], E-Shyong Tai[30,125,313], Alexander Teumer [131,314], Nicholas J Timpson [96,315,316], Daniela Toniolo[215], David-Alexandre Tregouet[41], Tiinamaija Tuomi [10,317,318,319], Peter Vollenweider[320], Carol A Wang [297,298], David R Weir [268], John B Whitfield [43], Cisca Wijmenga[14], Tien-Yin Wong[111,112], John Wright[6], Jingyun Yang[248,249], Lei Yu[248,249], Babette S Zemel[271,321], Alan B Zonderman[86], Markus Perola[322], Patrik K.E. Magnusson [24], André G Uitterlinden [21,22,23], Jaspal S Kooner[20,40,258,323], Daniel I Chasman[18,209], Ruth J.F. Loos [16,324], Nora Franceschini[15], Lude Franke [14], Chris S Haley [236,325], Caroline Hayward [236], Robin G Walters [11],

John R.B. Perry[12], Tõnu Esko[110,326], Agnar Helgason[5,133], Kari Stefansson[5,82], Peter K Joshi [1],
Michiaki Kubo[182] & James F Wilson [1,236]*

[1]Centre for Global Health Research, Usher Institute, University of Edinburgh, Edinburgh EH8 9AG, Scotland. [2]Department of Statistical Genetics, Osaka University Graduate School of Medicine, Suita, Osaka 565-0871, Japan. [3]Laboratory for Statistical Analysis, RIKEN Center for Integrative Medical Sciences, Yokohama, Kanagawa 230-0045, Japan. [4]Laboratory of Statistical Immunology, Immunology Frontier Research Center (WPI-IFReC), Osaka University, Suita, Osaka 565-0871, Japan. [5]deCODE genetics/Amgen Inc., Reykjavik 101, Iceland. [6]Bradford Institute for Health Research, Bradford Teaching Hospitals NHS Trust, Bradford BD96RJ, UK. [7]Research Unit, Area Science Park, Trieste 34149, Italy. [8]Department of Medicine, Surgery and Health Sciences, University of Trieste, Trieste, Italy. [9]Unit of Public Health Solutions, National Institute for Health and Welfare, Helsinki, Finland. [10]Institute for Molecular Medicine Finland, University of Helsinki, Helsinki, Finland. [11]Nuffield Department of Population Health, University of Oxford, Oxford OX3 7LF, UK. [12]MRC Epidemiology Unit, University of Cambridge School of Clinical Medicine, Cambridge CB2 0QQ, UK. [13]Cardiovascular Epidemiology Unit, Department of Public health and Primary Care, University of Cambridge, Cambridge CB1 8RN, UK. [14]Department of Genetics, University Medical Centre Groningen, University of Groningen, Groningen, the Netherlands, Groningen, Groningen 9700 RB, The Netherlands. [15]Department of Epidemiology, Gillings School of Global Public Health, University of North Carolina, Chapel Hill, NC 27514, USA. [16]The Charles Bronfman Institute for Personalized Medicine, Ichan School of Medicine at Mount Sinai, New York, NY 10029, USA. [17]Division of Genomic Outcomes, Department of Pediatrics, The Institute for Translational Genomics and Population Sciences, LABioMed at Harbor-UCLA Medical Center, Torrance, California 90502, USA. [18]Division of Preventive Medicine, Brigham and Women's Hospital, Boston, MA 02215, USA. [19]Department of Epidemiology and Biostatistics, Imperial College London, London W2 1PG, UK. [20]Department of Cardiology, Ealing Hospital, Middlesex, Middlesex UB1 3HW, UK. [21]Department of Internal Medicine, Erasmus University Medical Center, Rotterdam 3015 CN, Netherlands. [22]Department of Epidemiology, Erasmus University Medical Center, Rotterdam 3015 CN, Netherlands. [23]The Generation R Study Group, Erasmus University Medical Center, Rotterdam 3015 CN, The Netherlands. [24]Department of Medical Epidemiology and Biostatistics, Karolinska Institutet, Stockholm 17177, Sweden. [25]Institute for Social and Economic Research, University of Essex, Colchester CO4 3SQ, UK. [26]Cardiovascular Health Research Unit, Departments of Biostatistics and Medicine, University of Washington, Seattle, WA 98101, USA. [27]Research Unit of Molecular Epidemiology, Institute of Epidemiology, Helmholtz Zentrum München - German Research Center for Environmental Health, Neuherberg 85764, Germany. [28]Institute of Molecular Genetics, National Research Council of Italy, Pavia 27100, Italy. [29]Department of Biochemistry, University of Otago, Dunedin 9054, New Zealand. [30]Saw Swee Hock School of Public Health, National University of Singapore, Singapore, Singapore 117549, Singapore. [31]Department of Computational Biology, University of Lausanne, Lausanne 1011, Switzerland. [32]Center for Primary Care and Public Health (Unisanté), University of Lausanne, Lausanne, Switzerland. [33]Swiss Institute of Bioinformatics, Lausanne 1015, Switzerland. [34]Division of Human Genetics, Children's Hospital of Philadelphia, Philadelphia, PA 19104, USA. [35]Department of Genetics, Perelman School of Medicine, University of Pennsylvania, Philadelphia, PA 19104, USA. [36]Department of Neurology, Brain Centre Rudolf Magnus, University Medical Centre Utrecht, Utrecht University, Utrecht 3584 CX, The Netherlands. [37]Genomics Research Centre, School of Biomedical Sciences, Institute of Health and Biomedical Innovation, Queensland University of Technology, Brisbane, Queensland 4059, Australia. [38]Malaghan Institute of Medical Research, Wellington 6242, New Zealand. [39]Institute for Biomedicine, Eurac Research, Affiliated Institute of the University of Lübeck, Bolzano 39100, Italy. [40]MRC-PHE Centre for Environment and Health, Imperial College London, London W2 1PG, UK. [41]INSERM UMR_S 1166, Sorbonne Universités, Paris 75013, France. [42]ICAN Institute for Cardiometabolism and Nutrition, Paris 75013, France. [43]QIMR Berghofer Institute of Medical Research, Brisbane, Australia. [44]Department of Clinical Epidemiology, Leiden University Medical Center, Leiden 2333 ZA, The Netherlands. [45]Centre for Cognitive Ageing and Cognitive Epidemiology, University of Edinburgh, Edinburgh EH8 9JZ, UK. [46]Centre for Genomic & Experimental Medicine, Institute of Genetics & Molecular Medicine, University of Edinburgh, Edinburgh EH4 2XU, UK. [47]Clinical Division of Neurogeriatrics, Department of Neurology, Medical University of Graz, Graz 8036, Austria. [48]Institute of Medical Informatics, Statistics and Documentation, Medical University of Graz, Graz 8036, Austria. [49]CONACyT, Instituto Nacional de Ciencias Médicas y Nutrición Salvador Zubirán, Mexico 03940, México. [50]Department of Human Genetics, University of Chicago, Chicago, IL 60637, USA. [51]Department of Complex Trait Genetics, Center for Neurogenomics and Cognitive Research, Vrije Universiteit Amsterdam, Amsterdam 1081 HV, The Netherlands. [52]Alzheimer Center Department of Neurology, VU University Medical Center, Amsterdam Neuroscience, Amsterdam 1081HV, The Netherlands. [53]Interfaculty Institute for Genetics and Functional Genomics, University Medicine Greifswald, Greifswald 17475, Germany. [54]Chair of Experimental Bioinformatics, TUM School of Life Sciences Weihenstephan, Technical University of Munich, Freising-Weihenstephan 85354, Germany. [55]Department of Immunology, Genetics and Pathology, Science for Life Laboratory, Uppsala University, 75108 Uppsala, Sweden. [56]Vth Department of Medicine (Nephrology, Hypertensiology, Rheumatology, Endocrinology, Diabetology), Medical Faculty Mannheim, Heidelberg University, Mannheim 68167, Germany. [57]Cancer Genomics Research Laboratory, Leidos Biomedical Research, Inc., Frederick National Lab for Cancer Research, Frederick, MD, USA. [58]Wellcome Centre for Human Genetics, University of Oxford, Oxford OX3 7BN, UK. [59]Department of Computational Biology and Medical Sciences, Graduate school of Frontier Sciences, The University of Tokyo, Tokyo 108-8639, Japan. [60]Department of Molecular Epidemiology, German Institute of Human Nutrition Potsdam-Rehbruecke, Nuthetal, Germany. [61]German Center for Diabetes Research (DZD), München-Neuherberg, Germany. [62]Medical Research Institute, Ninewells Hospital and School of Medicine, University of Dundee, Dundee, UK. [63]Division of Endocrinology, Diabetes and Nutrition, Department of Medicine, University of Maryland, School of Medicine, Baltimore, MD 21201, USA. [64]Program for Personalized and Genomic Medicine, Department of Medicine, University of Maryland, School of Medicine, Baltimore, MD 21201, USA. [65]Department of Epidemiology, University of Groningen, University Medical Center Groningen, Groningen 9700 RB, The Netherlands. [66]Institute for Cardiogenetics, University of Lübeck, Lübeck 23562, Germany. [67]DZHK (German Research Centre for Cardiovascular Research), partner site Hamburg/Lübeck/Kiel, Lübeck 23562, Germany. [68]University Heart Center Luebeck, Lübeck 23562, Germany. [69]Charité – University Medicine Berlin, corporate member of Freie Universität Berlin, Humboldt-Universität zu Berlin, and Berlin Institute of Health, Institute for Dental and Craniofacial Sciences, Department of Periodontology and Synoptic Dentistry, Berlin, Germany. [70]Institute of Genetics and Biophysics A. Buzzati-Traverso - CNR, Naples 80131, Italy. [71]Finnish Institute for Molecular Medicine, University of Helsinki, Helsinki, Finland. [72]Genomics and Molecular Medicine Unit, CSIR-Institute of Genomics and Integrative Biology, New Delhi 110020, India. [73]Department of Clinical Sciences, Diabetes and Endocrinology, Lund University Diabetes Centre, Lund University, Skåne University Hospital, Malmö 20502, Sweden. [74]Genomic Research on Complex diseases (GRC Group), CSIR-Centre for Cellular and Molecular Biology, Hyderabad, Telangana 500007, India. [75]ePhood Scientific Unit, ePhood SRL, Bresso (Milano) 20091, Italy. [76]Department of Health Sciences, University of Milano, Milano 20139, Italy. [77]Neuroalgology Unit, IRCCS Foundation Carlo Besta Neurological Institute, Milano 20133, Italy. [78]Department of Pediatrics, College of Medicine, University of Oklahoma Health Sciences Center, Oklahoma City, OK 73104, USA. [79]Center for Research on Genomics and Global Health, National Human Genome Research Institute, National Institutes of Health, Bethesda, Maryland 20892-5635, USA. [80]Department of Epidemiology Research, Statens Serum Institut, Copenhagen DK-2300, Denmark. [81]Icelandic Heart Association, Kopavogur 201, Iceland. [82]Faculty of Medicine, School of Health Sciences, University of Iceland,

Reykjavik 101, Iceland. [83]Department of Genetics, University of North Carolina, Chapel Hill, NC 27599, USA. [84]Institute of Health and Wellbeing, University of Glasgow, Glasgow G12 8RZ, UK. [85]Cardiovascular Medicine Unit, Department of Medicine Solna, Centre for Molecular Medicine, Karolinska Institutet, Stockholm 171 76, Sweden. [86]Laboratory of Epidemiology and Population Sciences, National Institute on Aging, National Institutes of Health, Baltimore City, Maryland 21224, USA. [87]Department of Cardiology, Leiden University Medical Center, Leiden 2300 RC, the Netherlands. [88]Section of Gerontology and Geriatrics, Department of Internal Medicine, Leiden University Medical Center, Leiden 2300RC, the Netherlands. [89]Department of Epidemiology, Harvard T.H. Chan School of Public Health, Boston, MA 02115, USA. [90]Program in Genetic Epidemiology and Statistical Genetics, Harvard T.H. Chan School of Public Health, Boston, MA 02115, USA. [91]University of Groningen, University Medical Center Groningen, Department of Cardiology, Ther Netherlands, Groningen 9713 GZ, the Netherlands. [92]Italian Institute for Genomic Medicine (IIGM) and Dept. Medical Sciences, University of Turin, Italy, Turin 10126, Italy. [93]Division of Statistical Genomics, Department of Genetics, Washington University School of Medicine, Saint Louis, MO 63110-1093, USA. [94]NIHR Barts Cardiovascular Biomedical Research Centre, Barts and The London School of Medicine and Dentistry, Queen Mary University of London, London EC1M 6BQ, UK. [95]Department of Clinical Pharmacology, William Harvey Research Institute, Barts and The London School of Medicine and Dentistry, Queen Mary University of London, London EC1M 6BQ, UK. [96]MRC Integrative Epidemiology Unit at the University of Bristol, Bristol BS8 2BN, UK. [97]School of Psychological Science, University of Bristol, Bristol BS8 1TU, UK. [98]Department of Medicine, GeneSTAR Research Program, Johns Hopkins University School of Medicine, Baltimore, MD 21287, USA. [99]Department of Genetic Medicine, Weill Cornell Medicine Qatar, Doha, Qatar. [100]Computer and Systems Engineering, Alexandria University, Alexandria, Egypt. [101]Department of Epidemiology, School of Public Health, University of Michigan, Ann Arbor, MI 48109, USA. [102]Departamento de Endocrinología y Metabolismo, Instituto Nacional de Ciencias Médicas y Nutrición Salvador Zubirán, Mexico 14080, México. [103]Unidad de Investigacion de Enfermades Metabolicas, Tecnologico de Monterrey, Escuela de Medicina y Ciencias de la Salud, Monterrey, N.L. 64710, México. [104]Department of Ophthalmology, Graduate School of Medical Sciences, Kyushu University, Fukuoka, Fukuoka 812-8582, Japan. [105]Immunobiology of Dendritic Cells, Institut Pasteur, Paris 75015, France. [106]Inserm U1223, Paris 75015, France. [107]Centre for Translational Research, Institut Pasteur, Paris 75015, France. [108]Department of Cancer Immunology, Genentech Inc, San Francisco, California 94080, USA. [109]Division of Preventive Medicine, Department of Family Medicine and Public Health, UC San Diego School of Medicine, La Jolla, California 92093, USA. [110]Estonian Genome Center, University of Tartu, University of Tartu, Tartu 51010, Estonia. [111]Singapore Eye Research Institute, Singapore National Eye Centre, Singapore, Singapore 169856, Singapore. [112]Ophthalmology & Visual Sciences Academic Clinical Program (Eye ACP), Duke-NUS Medical School, Singapore, Singapore 169857, Singapore. [113]Department of Ophthalmology, Yong Loo Lin School of Medicine, National University of Singapore, Singapore, Singapore 119228 SG, Singapore. [114]Endocrine Research Center, Research Institute for Endocrine Sciences, Shahid Beheshti University of Medical Sciences, Tehran 19839-63113, Iran. [115]Department of Epidemiology, German Institute of Human Nutrition Potsdam-Rehbruecke, Nuthetal, Germany. [116]Health Science Center at Houston, UTHealth School of Public Health, University of Texas, Houston, TX 77030, USA. [117]USC-Office of Population Studies Foundation, Inc., Department of Nutrition and Dietetics, Talamban, University of San Carlos, Cebu City 6000 Cebu, Philippines. [118]Department of Vascular Surgery, Division of Surgical Specialties, University Medical Center Utrecht, University of Utrecht, Utrecht, Utrecht 3584 CX, Netherlands. [119]Digital Health Center, Hasso Plattner Institute, Universität Potsdam, Potsdam 14482, Germany. [120]Division of Cancer Epidemiology & Genetics, National Cancer Institute, National Institutes of Health, Bethesda, MD 20892, USA. [121]Institute of Population Health Sciences, National Health Research Institutes, Miaoli, Taiwan, Taiwan. [122]Institute for Maternal and Child Health - IRCCS Burlo Garofolo, Trieste 34137, Italy. [123]National Human Genome Research Institute, National Institutes of Health, Bethesda, Maryland 20892, USA. [124]Department of Psychology, University of Edinburgh, 7 George Square, Edinburgh EH8 9JZ, UK. [125]Department of Medicine, Yong Loo Lin School of Medicine, National University of Singapore, Singapore 119228 SG, Singapore. [126]Department of Nutrition, Harvard T.H. Chan School of Public Health, Boston, Massachusetts 02115, USA. [127]Cellular and Molecular Endocrine Research Center, Research Institute for Endocrine Sciences, Shahid Beheshti University of Medical Sciences, Tehran 19839-63113, Iran. [128]Department of Internal Medicine, Rush University Medical Center, Chicago, Illinois, USA. [129]MEMO Research, Molecular and Clinical Medicine, University of Dundee, Dundee DD19SY, UK. [130]Department of Internal Medicine B, University Medicine Greifswald, Greifswald 17475, Germany. [131]DZHK (German Centre for Cardiovascular Research), partner site Greifswald, Greifswald 17475, Germany. [132]Department of Genetics, King Faisal Specialist Hospital and Research Center, Riyadh, KSA 12713, Saudi Arabia. [133]Department of Anthropology, University of Iceland, Reykjavik 101, Iceland. [134]National Institute for Health Research Imperial Biomedical Research Centre, Imperial College Healthcare NHS Trust and Imperial College London, London, UK. [135]UK Dementia Research Institute (UK DRI) at Imperial College London, London, UK. [136]Health Data Research UK - London, London, England. [137]Department of Pediatrics, Erasmus University Medical Center, Rotterdam 3015CN, The Netherlands. [138]Section on Nephrology, Department of Internal Medicine, Wake Forest School of Medicine, Winston-Salem, NC 27101, US. [139]Medical Genetics, Institute for Maternal and Child Health - IRCCS Burlo Garofolo, Trieste, Italy. [140]Division of Cardiovascular Medicine, Radcliffe Department of Medicine, University of Oxford, Oxford OX3 9DU, UK. [141]Division of Endocrinology, Diabetes, and Metabolism, Department of Medicine, Cedars-Sinai Medical Center, Los Angeles, California 90048, USA. [142]Braun School of Public Health, Hebrew University-Hadassah Medical Center, Jerusalem, Israel. [143]School of Engineering and Natural Sciences, University of Iceland, Reykjavik 101, Iceland. [144]Chinese Academy of Medical Sciences, Beijing 100730, China. [145]Social, Genetic and Developmental Psychiatry Centre, Institute of Psychiatry, Psychology & Neuroscience, King's College London, London SE5 8AF, UK. [146]Department of Preventive Medicine, Keck School of Medicine, University of Southern California, Los Angeles, California 90089, USA. [147]Blizard Institute, Queen Mary University of London, London E1 2AT, UK. [148]Laboratory of Genome Technology, Institute of Medical Science, The University of Tokyo, Tokyo 108-8639, Japan. [149]Laboratory of Clinical Chemistry and Hematology, Division Laboratories and Pharmacy, University Medical Center Utrecht, University of Utrecht, Utrecht, Utrecht 3584 CX, Netherlands. [150]Shanghai Institute of Hematology, State Key Laboratory Of Medical Genomics, Rui-jin Hospital, Shanghai Jiao Tong University School of Medicine, Shanghai, China 200025, China. [151]Division of Endocrine and Metabolism, Tri-Service General Hospital Songshan branch, Taipei, Taiwan, Taiwan. [152]School of Medicine, National Defense Medical Center, Taipei, Taiwan, Taiwan. [153]Unit of Public Health Promotion, National Institute for Health and Welfare, Helsinki, Finland. [154]Center for Genomic Medicine, Kyoto University Graduate School of Medicine, Kyoto 606-8507, Japan. [155]Department of Biomedical Informatics, Harvard Medical School, Boston, MA 02115, USA. [156]Deutsches Herzzentrum München, Klinik für Herz- und Kreislauferkrankungen, Technische Universität München, Munich 80636, Germany. [157]Department of Public Health and Primary Care, University of Cambridge, Cambridge CB2 0SR, UK. [158]Human Genetics, Genome Institute of Singapore, Agency for Science, Technology and Research, Singapore, Singapore 138672, Singapore. [159]Health Services and Systems Research, Duke-NUS Medical School, Singapore, Singapore 169857. [160]Centre for Global Health, Faculty of Medicine, University of Split, Split, Croatia. [161]Institute of Clinical Medicine, Internal Medicine, University of Eastern Finland, Kuopio, Finland. [162]Kuopio University Hospital, Kuopio, Finland. [163]Bristol NIHR Biomedical Research Centre, Bristol BS8 2BN, UK. [164]Population Health Science, Bristol Medical School, Bristol BS8 2BY, UK. [165]Division of Endocrinology and Metabolism, Department of Internal Medicine, Taichung Veterans General Hospital, Taichung, Taiwan, Taiwan. [166]School of Medicine, National Yang-Ming University, Taipei, Taiwan, Taipei 112, Taiwan. [167]School of Medicine, Chung Shan Medical University, Taichung, Taiwan, Taichung City 402, Taiwan. [168]Department of Medical Research, Taichung Veterans General Hospital, Taichung, Taiwan, Taiwan. [169]Department of Internal Medicine A, University Medicine Greifswald, Greifswald 17475, Germany. [170]Department of Epidemiology and Biostatistics, Peking University

Health Science Centre, Peking University, Beijing 100191, China. [171]Translational Laboratory in Genetic Medicine, Agency for Science, Technology and Research, Singapore (A*STAR), Singapore 138648, Singapore. [172]Department of Biochemistry, Yong Loo Lin School of Medicine, National University of Singapore, Singapore, Singapore 117596, Singapore. [173]National Institute of Environmental Health Sciences, National Institutes of Health, Department of Health and Human Services, Research Triangle Park, Durham, NC 27709, USA. [174]Department of Epidemiology, Johns Hopkins Bloomberg School of Public Health, Baltimore, MD 21205, USA. [175]Center for Public Health Genomics, University of Virginia School of Medicine, Charlottesville, VA 22908, USA. [176]Genomics of Renal Diseases and Hypertension Unit, IRCCS San Raffaele Scientific Institute, Università Vita Salute San Raffaele, Milano 20132, Italy. [177]Helmholtz Zentrum München, Independent Research Group Clinical Epidemiology, Neuherberg 85764, Germany. [178]Institute of Human Genetics, Helmholtz Zentrum Muenchen, Neuherberg 85764, Germany. [179]Institute of Human Genetics, Technical University of Munich, Munich 81675, Germany. [180]DZHK (German Center for Cardiovascular Research), partner site Munich Heart Alliance, Munich 80802, Germany. [181]Estonian Genome Center, Institute of Genomics, University of Tartu, Tartu 51010, Estonia. [182]Laboratory for Genotyping Development, RIKEN Center for Integrative Medical Sciences, Yokohama, Kanagawa 230-0045, Japan. [183]Laboratory of Haematology, La Timone Hospital, Marseille, France. [184]INSERM UMR_S 1263, Center for CardioVascular and Nutrition research (C2VN), Aix-Marseille University, Marseille, France. [185]Departamento de Economía, Universidad Autónoma Metropolitana, Mexico 09340, México. [186]Medical School, The University of Western Australia, Perth, Western Australia/Australia 6009, Australia. [187]The University of Texas Health Science Center at Houston, School of Public Health, Department of Epidemiology, Human Genetics and Environmental Sciences, Houston, Texas 77030, USA. [188]Institute of Social and Preventive Medicine (ISPM), University of Bern, Bern, Switzerland. [189]Division of Molecular Pathology, Institute of Medical Science, The University of Tokyo, Tokyo 108-8639, Japan. [190]The Institute of Medical Sciences, Aberdeen Biomedical Imaging Centre, University of Aberdeen, Aberdeen AB25 2ZD, UK. [191]Laboratory of Neurogenetics, Bethesda, MD 20892, USA. [192]Data Tecnica International LLC, Glen Echo, MD 20812, USA. [193]Institute of Clinical Chemistry and Laboratory Medicine, University Medicine Greifswald, Greifswald 17475, Germany. [194]Oxford Centre for Diabetes, Endocrinology and Metabolism, University of Oxford, Headington, Oxford OX3 7LJ, UK. [195]Oxford NIHR Biomedical Research Centre, Oxford University Hospitals Trust, Oxford, UK. [196]Department of Paediatrics, University of Cambridge School of Clinical Medicine, Cambridge CB2 0QQ, UK. [197]Instituto Nacional de Medicina Genómica, Mexico 14610, México. [198]Institute of Cardiovascular and Medical Sciences, University of Glasgow, Glasgow G12 8TA, UK. [199]Division of Epidemiology & Community Health, School of Public Health, University of Minnesota, Minneapolis, MN 55454, USA. [200]Gen-info Ltd, Zagreb, Croatia, Zagreb, Select a Province 10000, Croatia. [201]International Centre for Circulatory Health, Imperial College London, London W2 1PG, UK. [202]Imperial Clinical Trials Unit, Imperial College London, London, London W12 7TA, UK. [203]Human Evolutionary Genetics Unit, Institut Pasteur, Paris 75015, France. [204]Centre National de la Recherche Scientifique (CNRS) UMR2000, Paris 75015, France. [205]Center of Bioinformatics, Biostatistics and Integrative Biology, Institut Pasteur, Paris 75015, France. [206]Department of Psychology and Logopedics, Faculty of Medicine, University of Helsinki, University of Helsinki, Helsinki 00014, Finland. [207]Hero Heart Institute and Dyanand Medical College and Hospital, Ludhiana, Punjab, India. [208]Division of Biostatistics, Washington University School of Medicine, St. Louis, Missouri, USA. [209]Harvard Medical School, Boston, MA 02115, USA. [210]Department of Applied Economics, Erasmus School of Economics, Erasmus University Rotterdam, Rotterdam 3062 PA, The Netherlands. [211]Erasmus University Rotterdam Institute for Behavior and Biology, Erasmus University Rotterdam, Rotterdam 3062 PA, The Netherlands. [212]IRCCS Neuromed, Pozzilli (IS) 86077, Italy. [213]Gottfried Schatz Research Center (for Cell Signaling, Metabolism and Aging), Division of Molecular Biology and Biochemistry, Medical University of Graz, 8010 Graz, Austria. [214]Unit of Genomics of Complex Diseases, Institut de Recerca Hospital de la Santa Creu i Sant Pau, IIB-Sant Pau, Barcelona, Spain. [215]San Raffaele Research Institute, Milano, Italy. [216]Department of Public Health Solutions, National Institute for Health and Welfare, Helsinki FI-00271, Finland. [217]Department of Biostatistics, and Center for Statistical Genetics, University of Michigan, Ann Arbor, Michigan 48109, USA. [218]Dept of Cell and Molecular Biology, National Bioinformatics Infrastructure Sweden, Science for Life Laboratory, Uppsala University, Uppsala, SE-752 37 Uppsala, Sweden. [219]Department of Cardiology, Division Heart & Lungs, University Medical Center Utrecht, University of Utrecht, Utrecht, Utrecht 3485 CX, Netherlands. [220]School of Medicine, National Yang-Ming University, Taipei, Taiwan, Taiwan. [221]Institute of Medical Technology, National Chung-Hsing University, Taichung, Taiwan, Taiwan. [222]School of Basic and Applied Sciences, Dayananda Sagar University, Bangalore, Karnataka 560078, India. [223]Diabetes Unit, KEM Hospital and Research Centre, Pune, Maharashtra 411101, India. [224]Symbiosis Statistical Institute, Symbiosis International University, Pune, Maharashtra 411007, India. [225]Panjab University, Chandigarh, India. [226]Division of Population Health Sciences, Ninewells Hospital and Medical School, University of Dundee, Dundee DD1 9SY, UK. [227]RCSI Molecular & Cellular Therapeutics (MCT), Royal College of Surgeons in Ireland, RCSI Education & Research Centre, Beaumont Hospital, Dublin 9, Ireland. [228]Alzheimer Scotland Dementia Research Centre, University of Edinburgh, Edinburgh EH8 9JZ, Scotland. [229]School of Physiotherapy and Exercise Science, Faculty of Health Sciences, Curtin University, Perth, Western Australia/Australia 6102, Australia. [230]Department of Respiratory Medicine, Ghent University Hospital, Ghent 9000, Belgium. [231]Department of Pathology, University of Vermont, Colchester, VT 05446, USA. [232]Departamento de Medicina Genómica y Toxicología Ambiental, Instituto de Investigaciones Biomédicas, UNAM, Mexico 04510, México. [233]Unidad De Biología Molecular y Medicina Genómica, Instituto Nacional de Ciencias Médicas y Nutrición Salvador Zubirán, Mexico 14080, México. [234]Department of Hygiene and Epidemiology, University of Ioannina Medical School, Ioannina 45110, Greece. [235]Institute of Genetic and Biomedical Research - Support Unity, National Research Council of Italy, Rome, Italy. [236]MRC Human Genetics Unit, Institute of Genetics and Molecular Medicine, University of Edinburgh, Edinburgh EH4 2XU, Scotland. [237]Institute for Molecular Medicine Finland (FIMM), University of Helsinki, Helsinki FI-00014, Finland. [238]Department of Cardiology, Hero DMC Heart Institute, Dayanand Medical College & Hospital, Ludhiana, Punjab 141001, India. [239]Beijing Institute of Ophthalmology, Beijing Tongren Eye Center, Beijing Tongren Hospital, Capital Medical University, Beijing Ophthalmology and Visual Science Key Lab, Beijing, China 100005, China. [240]Centre for Population Health Sciences_Usher Institute of Population Health and Informatics, University of Edinburgh, Edinburgh EH8 9AG, Scotland. [241]Diabetes Unit, K.E.M. Hospital Research Centre, Pune, MAH 411011, India. [242]Department of Epidemiology, Graduate School of Public Health, University of Pittsburgh, Pittsburgh, Pennsylvania, USA. [243]Department of Cardiology, Division Heart & Lungs, University Medical Center Utrecht, University of Utrecht, Utrecht, Utrecht 3584 CX, Netherlands. [244]Institute of Cardiovascular Science, Faculty of Population Health Sciences, University College London, London WC1E 6DD, UK. [245]Durrer Center for Cardiovascular Research, Netherlands Heart Institute, Utrecht, Netherlands. [246]Farr Institute of Health Informatics Research and Institute of Health Informatics, University College London, London, UK. [247]Department of Internal Medicine, University Medical Center Groningen, University of Groningen, Groningen 9713GZ, The Netherlands. [248]Rush Alzheimer's Disease Center, Rush University Medical Center, Chicago, IL 60612, USA. [249]Department of Neurological Sciences, Rush University Medical Center, Chicago, IL 60612, USA. [250]Systems Genomics Laboratory, School of Biotechnology, Jawaharlal Nehru University, New Delhi 110067, India. [251]Pennington Biomedical Research Center, Baton Rouge, Louisiane 70808, USA. [252]Center for Applied Genomics, Division of Human Genetics, Children's Hospital of Philadelphia, Philadelphia, PA 19104, USA. [253]Quantinuum Research LLC, San Diego, CA 92101, USA. [254]Cardiovascular Health Research Unit, Department of Medicine, University of Washington, Seattle, WA 98101, USA. [255]Center for Experimental Social Science, Department of Economics, New York University, New York, New York 10012, USA. [256]Research Institute for Industrial Economics (IFN), Stockholm 102 15, Sweden. [257]Lee Kong Chian School of Medicine, Nanyang Technological University, Singapore 308232, Singapore. [258]Imperial College Healthcare NHS Trust, London, London W12 0HS, UK. [259]Department of Preventive Medicine, Northwestern University

Feinberg School of Medicine, Chicago, IL 60611, USA. [260]Institute of Biomedical Technologies Milano, National Research Council of Italy (CNR), Segrate (Milano) 20090, Italy. [261]Bio4Dreams Scientific Unit, Bio4Dreams SRL, Bio4Dreams - business nursery for life sciences, Milano 20121, Italy. [262]Institute of Clinical Molecular Biology, Christian-Albrechts-University of Kiel, 24105 Kiel, Germany. [263]Department of General Practice and Primary health Care, University of Helsinki, Tukholmankatu 8 B, Helsinki 00014, Finland. [264]National Institute for Health and Welfare, Helsinki, Finland. [265]Unit of General Practice, Helsinki University Central Hospital, Helsinki, Finland. [266]Folkhälsan Research Centre, Helsinki, Finland. [267]Vasa Central Hospital, Vaasa, Finland. [268]Survey Research Center, Institute for Social Research, University of Michigan, Ann Arbor, MI 48014, USA. [269]Montreal Heart Institute, Montreal, QC, Canada. [270]Centro de Estudios en Diabetes, Unidad de Investigacion en Diabetes y Riesgo Cardiovascular, Centro de Investigacion en Salud Poblacional, Instituto Nacional de Salud Publica, Cuernavaca 01120, México. [271]Department of Pediatrics, Perelman School of Medicine, University of Pennsylvania, Philadelphia, PA 19104, USA. [272]Cardiovascular Medicine Unit, Department of Medicine Solna, Centre for Molecular Medicine, Stockholm 171 76, Sweden. [273]Department of Paediatrics, Yong Loo Lin School of Medicine, National University of Singapore, Singapore, Singapore. [274]Khoo Teck Puat - National University Children's Medical Institute, National University Health System, Singapore, Singapore. [275]Research Unit of Internal Medicine, Medical Research Center Oulu, University of Oulu and Oulu University Hospital, Oulu 90014, Finland. [276]Division of Epidemiology, Department of Internal Medicine, University of Utah School of Medicine, Salt Lake City, Utah 84108, USA. [277]Center for Translational & Computational Neuroimmunology, Department of Neurology, Columbia University Medical Center, 650 West 168th street, PH19-311, Newyork, NY 10032, USA. [278]Cell Circuits Program, Broad Institute, Cambridge, MA 02142, USA. [279]Department of Economics, Stockholm School of Economics, Stockholm SE-113 83, Sweden. [280]Department of Ophthalmology, Medical Faculty Mannheim of the Ruprecht-Karls-University of Heidelberg, Mannheim 698167, Germany. [281]Department of Public Health, University of Helsinki, Helsinki FI-00014, Finland. [282]Oxford NIHR Biomedical Research Centre, Oxford University Hospitals NHS Foundation Trust, John Radcliffe Hospital, Oxford OX3 9DU, UK. [283]Laboratory of Clinical Chemistry and Hematology, Division Laboratories and Pharmacy, University Medical Center Utrecht, University of Utrecht, Utrecht 3584 CX, Netherlands. [284]Helsinki Collegium for Advanced Studies, University of Helsinki, University of Helsinki, Helsinki 00014, Finland. [285]University Hospital Schleswig-Holstein (UKSH), Campus Kiel, Kiel 24105, Germany. [286]Institute of Epidemiology and PopGen Biobank, University of Kiel, Kiel, Schleswig Holstein 24105, Germany. [287]Department of Statistics, University of Auckland, Auckland, New Zealand. [288]Clinical Institute of Medical and Chemical Laboratory Diagnostics, Medical University of Graz, Graz, Austria. [289]Synlab Academy, Synlab Holding Deutschland GmbH, Mannheim, Germany. [290]Department of Genetics, King Faisal Specialist Hospital and Research Centre, Riyadh 11211, Saudi Arabia. [291]Institute for Molecular Bioscience, The University of Queensland, Brisbane, Queensland 4072, Australia. [292]Department of Public Health and Primary Care, Leiden University Medical Center, Leiden 2333 ZA, The Netherlands. [293]School of Biomedical Sciences, Institute of Health and Biomedical Innovation, Queensland University of Technology, Kelvin Grove, QLD 4059, Australia. [294]Department of Psychiatry, Interdisciplinary Center Psychopathology and Emotion Regulation, University of Groningen, University Medical Center Groningen, Groningen 9700 RB, The Netherlands. [295]Department of Medicine, Columbia University Medical Center, New York, New York, USA. [296]Pat Macpherson Centre for Pharmacogenetics and Pharmacogenomics, The School of Medicine, University of Dundee, Dundee DD1 9SY, UK. [297]School of Medicine and Public Health, Faculty of Medicine and Health, The University of Newcastle, Newcastle, New South Wales, Australia. [298]Division of Obstetrics and Gynaecology, The University of Western Australia, Perth, Western Australia/Australia 6009, Australia. [299]Department of kinesiology, Laval University, Quebec, QC G1V 0A6, Canada. [300]Institute of Nutrition and Functional Foods, Laval University, Quebec, QC G1V 0A6, Canada. [301]Institute of Genetic and Biomedical Research - Support Unity, National Research Council of Italy, Sassari 07100, Italy. [302]Department of Clinical Genetics, Amsterdam Neuroscience, VU Medical Centre, Amsterdam 1081HV, The Netherlands. [303]Cardiovascular Health Research Unit, Departments of Epidemiology, Medicine and Health Services, University of Washington, Seattle, WA 98101, USA. [304]Kaiser Permanente Washington Health Research Institute, Seattle, WA 98101, USA. [305]Department of Medicine, Faculty of Medicine, Université de Montréal, Montreal, Quebec H3T 1J4, Canada. [306]Laboratory of Experimental Cardiology, Division Heart & Lungs, University Medical Center Utrecht, University of Utrecht, Utrecht, Utrecht 3584 CX, Netherlands. [307]Oklahoma Center for Neuroscience, Oklahoma City, OK 73104, USA. [308]Institute of Nutritional Sciences, University of Potsdam, Nuthetal, Germany. [309]Deutsches Zentrum für Herz- und Kreislauferkrankungen (DZHK), Munich Heart Alliance, Munich 80636, Germany. [310]Regeneron Genetics Center, Regeneron Pharmaceuticals, Inc, Tarrytown, NY 10591-6607, USA. [311]Faculty of Health Studies, University of Bradford, Bradford, West Yorkshire BD7 1DP, UK. [312]Cardiovascular Health Research Unit, Division of Cardiology, University of Washington, Seattle, WA 98101, USA. [313]Duke-NUS Medical School, National University of Singapore, Singapore, Singapore 169857 SG, Singapore. [314]Institute for Community Medicine, University Medicine Greifswald, Greifswald 17475, Germany. [315]Department of Population Health Sciences, Bristol Medical School, University of Bristol, Bristol BS8 2PR, UK. [316]Avon Longitudinal Study of Parents and Children (ALSPAC), University of Bristol, Bristol BS8 2PR, UK. [317]Endocrinology, Abdominal Centre, University of Helsinki, Helsinki University Hospital, Helsinki, Finland. [318]Folkhalsan Research Center, Helsinki, Finland. [319]Research Program of Diabetes and Endocrinology, University of Helsinki, Helsinki, Finland. [320]Department of Medicine, Internal Medicine, Lausanne University Hospital, Lausanne 1011, Switzerland. [321]Division of Gastroenterology, Hepatology and Nutrition, Children's Hospital of Philadelphia, Philadelphia, PA 19146, USA. [322]Unit of Genomics and Biomarkers, National Institute for Health and Welfare, Helsinki 00271, Finland. [323]National Heart and Lung Institute, Imperial College London, London W12 0NN, UK. [324]The Mindich Child Health and Development Institute, The Icahn School of Medicine at Mount Sinai, New York, NY 10029, USA. [325]Roslin Institute and Royal (Dick) School of Veterinary Studies, University of Edinburgh, Easter Bush, Midlothian EH25 9RG, Scotland. [326]Program in Medical and Population Genetics, Broad Institute, Broad Institute, Cambridge, MASSACHUSETTS 02142, USA. [327]Deceased: Yucheng Jia, John M. Starr.

