## [Peer Review File · Nature Communications]

Reviewers' comments:

Reviewer #1 (Remarks to the Author):

This manuscript provides results from a very-large meta-analysis of the association between runs of homozygosity and phenotypes in the largest sample of humans available to date^{1,2}. Significant findings are presented for many human traits. However, my main concern is that many of the results that are highlighted in the paper are a consequence of the metric of inbreeding used and thus do not appear to be robust to a different choice of metric. Rather than the authors presenting an unbiased presentation of the results from all methods and investigating these differences further with simulation study and theory to challenge previous findings they have simply, to a large degree, just buried any discrepancies in their presentation. This with a combination of emotive writing and the use of unscientific subjective language of “weak to moderate”, “relatively unbiased”, “surprisingly”, etc. leaves me quite concerned that the authors are pushing a particular dogma, rather than exploring their data in full detail. Finally, I also find their conclusion that inbreeding depression is associated through the action of rare recessive variants to be completely unsupported by the analyses they have conducted.

A recent study calls into question the use of ROH as a measure of inbreeding (Yengo et al. 2017 PNAS). The authors report that: “This attenuation is greatest in low homozygosity cohorts, suggesting that F_{ROH} is a better estimator of the true inbreeding coefficient (Fig 3a).” This is extremely weak evidence for this, if any evidence at all. This attenuation could be greatest in low homozygosity cohorts because of a wide-range of factors, including that F_{GRM} is a better metric in unrelated, outbred samples for instance. This manuscript needs to fully justify its choice of metric by exploring through theory and simulation exactly why and when F_{GRM} provides a worse measure of inbreeding than F_{ROH} because current evidence suggests that associations using F_{GRM} are less biased, with results attenuated as compared to ROH and this is exactly what is found in this study. At the very very least the authors should present the F_{GRM} results in the main text as the main results and highlight that many of their new and “surprising” findings completely disappear when using the alternative metric, which has been shown in extensive simulation study and theory to be less biased than the metric used here.

ROH is more susceptible to stratification biases. Linear “dosage” responses, as the authors state, are also expected if ROH reflects the degree of population stratification and the covariance of environmental confounding. Thus, I find this to be no evidence that these associations do not reflect other underlying causes.

F_{SNP} (which is also a bad metric) outside of F_{ROH} tells one nothing of the minor allele frequency distribution of the directional dominance. No evidence in this manuscript is presented for an excess of rare-variant effects. One would have to repeat the entire analysis dividing the measure into MAF groups...so F_{GRM} for rare versus common variants when both are fitted in the model. I note that rare variants are actually excluded from the ROH calling (which is fine), but it means that little can be said about the frequency distribution of the effects, leaving current statements as nothing more than conjecture.

In the replication studies, please present the replication effects and not the ratios as the ratio distort the interpretation, please let the reader have all of the information. This is in the supplementary but it's a question of unbiased presentation of the results. I note that the sibling pairs and the adopted individuals were not left out of the original meta-analysis, this would be nice to have unbiased replication of the effects.

Line 443: linear mixed models are commonly used for binary traits under a liability threshold model to control for relatedness, so there is no reason to not control for family structure here.

The sibling analysis is very good and the method is very nice. Why don't you do it for the other

metrics also? Also, please consider fitting the sibling IBD deviation from 0.5 of the pair to account for mendelian segregation as a whole and additive effects before testing for directional dominance.

There are spousal pairs in the UK Biobank so why not test for assortment at ROH?

Reviewer #2 (Remarks to the Author):

Re: Effects of autozygosity on a broad range of human phenotypes.

In the manuscript the author wish to determine a link between the lengths of autozygous regions in a subject to a number of phenotypic parameters which vary between those that are self-reported and physical measurements. The majority of the phenotypes also have a strong environmental component. Regarding the conclusions reached by authors, while I agree that increased inbreeding does have a negative impact on an individual, I have a major concern that the demonstrated effects on a number of the phenotypes are due societal pressure which limits a person's behaviour/options rather than an underlying mechanism where by a person's genetic makeup drives their behaviour.

A family's cultural background is a strong factor in the likelihood that they practice consanguineous unions and so the typical length of homozygous regions in family members. Similarly, a family's cultural background is a strong factor how they would perform/report on a large number of the parameters discussed in the paper. Consequently, I'm very concerned that the length of a person's homozygous regions is a marker for their cultural 'view point' and it is this that determines their phenotype/behaviour. For instance the age at which they first have sex, number of partners, amount of alcohol consumed, what they view as risk behaviour or how well they have aged, who they chose to have children with, etc. is strongly impacted by their cultural environment. Therefore, while the authors did adjust for ethnic origin and religion I think they need to more extensively explore any possible link between cultural 'view point' of the individuals and their phenotype. For instance Pakistani Muslims are not a homogeneous group with individuals originating from the north-west tribal regions more likely to be consanguineous and socially conservative than those from the large cities in the south, consequently population stratification may have a strong affect leading to the apparent linking of length of autozygous regions to a wide range of behaviours.

I would also like to know a little more on how they filtered homozygous regions that spanned regions that are poorly genotyped in many genotype microarrays. For instances did they discount homozygous runs that spanned the centromeres, particularly of chromosomes 17 and 19 where poor genotyping of regions flanking the centromere frequently suggests homozygosity. Since centromeres are not genotyped they don't contain heterozygous SNP which would terminate the extension of the apparent homozygous regions across the centromere. This can result in the false reporting of homozygous runs of over 5Mb per chromosome.

Reviewer #3 (Remarks to the Author):

This study examined the relationship between individual-level autozygosity and a wide range of complex traits, reporting that autozygosity is associated with 32 of 100 traits analyzed. Even though this result is not surprising, it is important because the analysis was based on a very large sample and a large number of traits, and it has carefully explored potential confounding factors by using adopted individuals and sib pairs. What is particularly impressive is that the results are consistent across diverse ancestral groups and across several demographic scenarios that could

increase autozygosity.

I have only a few minor comments.

1. The genetics concepts can be defined more systematically. The study used long ROH regions as a surrogate of autozygosity of recent origin. More precisely, the study used genotype data to identify unusually long regions of ROH, which cannot be explained by the expected length distribution of ROH runs in an ideal population, with a large size and no inbreeding in the recent and mid-distance past. Operationally the study used 1.5 Mb as the cutoff, and used the total length of such long-ROH segments in an individual to calculate his/her genomic fraction accounted for by these regions. In contrast, the genomewide heterozygosity rate contains both the long-ROH fraction calculated above and the excess amount of homozygous genotypes outside of long-ROH regions. I think the authors did the right analyses, but hope it can be explained even more clearly. In particular, whether the second measure is based on each population's own allele frequencies is hard to find, and I assume it is and it is somewhere in the Method section. The discrepancy between these two measures reflects the relative contribution of autozygosity from different time depths; that is, how much is from very recent inbreeding events and how much is from more distant events, including the situation loosely called small effective population size.

2. Some other technical details are important enough that I think they can be brought forth, from Method to the main section. The Result mentioned "231 homogeneous sub-cohort". Here "homogeneous" probably meant there is no discernible within-cohort stratification, but each could have its own level of autozygosity. It is not immediately clear if homogeneity is based on a formal test, such as patterns of F_{st} or HWE test statistics over all the loci. F_{IS} calculation is affected by the genotyping panel (whether it contains mainly common or rare variants); so it seems that we need to be told more about potential caveats if different SNP arrays were used. The study nicely shows that the comparison of the two autozygosity measures: long-ROH vs. all-the-rest, can divide the cohorts into at least three demographic scenarios. However, one of them, large F_{IS} and small F_{ROH} , is for non-homogeneous cohorts.

3. A reader who only skim through the main text may still wonder if the effect size – the slope in the regression between the trait and F_{ROH} – is highly variable among the populations or ancestral groups. Maybe I missed it. In any case this point can be provided more centrally.

4. Age effect has been "removed" by being treated as a covariate in the model. I wonder if the effect of autozygosity has an interaction effect with age, for some of the traits.

5. Some phrases can be defined more clearly. "genetically correlated" is one. "true inbreeding coefficient" is problematic because different measures emphasize different time depth, and the field has been handicapped for so long by dichotomizing the concept into very recent (consanguinity) and very distant (small population size). In the context of this work, F_{ROH} captures a greater span of the genome than F_{IS} . Whether 1% in F_{ROH} is truly more "risky" than 1% in F_{IS} is a profound question. Technically it's difficult to calculate a score like F_{IS} that is specific for a defined model, such as co-ancestry in 10-200 generations-but-not-more-recent. Here the question is related to harder questions such as, is a rare recessive variant more impactful when it is contained in a long ROH than in a shorter ROH? I think this paper brought many exciting questions for the future.

Response to reviews

We thank the reviewers for their comments on our manuscript and thank the editor for the chance to respond to them. We shall address the comments sequentially below.

Reviewer #1:

This manuscript provides results from a very-large meta-analysis of the association between runs of homozygosity and phenotypes in the largest sample of humans available to date. Significant findings are presented for many human traits. However, my main concern is that many of the results that are highlighted in the paper are a consequence of the metric of inbreeding used and thus do not appear to be robust to a different choice of metric.

As this is the main concern of Reviewer 1 we have addressed this point in detail in a new Supplementary Note 4. In summary, we believe that F_{ROH} **based on accurate ROH calling** is an appropriate metric of inbreeding. Nevertheless, we do not agree that our results “are a consequence of the metric of inbreeding”. Of the 32 traits for which we find a significant association with F_{ROH} , 26 also have a significant association with F_{GRM} . We acknowledge that the choice of inbreeding metric does affect the magnitudes and standard errors of the effect size estimates, and we believe the reasons for these differences are interesting and informative, but we justify our focus on F_{ROH} in Supplementary Note 4.

Rather than the authors presenting an unbiased presentation of the results from all methods and investigating these differences further with simulation study and theory to challenge previous findings they have simply, to a large degree, just buried any discrepancies in their presentation.

We believe the “previous findings” referred to here are those of Yengo et al. PNAS (2017)⁴². In particular, this study states that “FROH may overestimate Inbreeding Depression by 162%”. This statement appears to conflict with over a decade of publications justifying FROH as a metric of inbreeding. To better understand the cause of this discrepancy we have performed the investigations suggested by R1 in Supplementary Notes 4 and 5. In Supplementary Note 4 we show that the results in Yengo et al. were biased by applying parameters developed for SNP-chip genotypes to imputed dosages. Specifically, we believe the higher error rate of imputed genotypes caused Yengo et al. to identify only a fraction of autozygous segments, as illustrated in Supplementary Figure 14a, below.

Supplementary Figure 14a: Comparing ROH called from SNP array genotypes and imputed dosages. For a single high F_{ROH} individual, the locations of called ROH are compared for two methods. The method shown in blue calls ROH from SNP array genotypes using the parameters specified in Joshi et al. (2015). The method shown in red calls ROH from hard called imputed dosages following the method described in Yengo et al. (2017). The long ROH detected in SNP array genotypes, which are thought to be autozygous segments, are broken up in the imputed data method by the presence of miscalled heterozygotes.

This with a combination of emotive writing and the use of unscientific subjective language of “weak to moderate”, “relatively unbiased”, “surprisingly”, etc. leaves me quite concerned that the authors are pushing a particular dogma, rather than exploring their data in full detail.

It was certainly not our intention to push a particular dogma, and we have now reworded the text in various places to alleviate the concerns of the reviewer.

Finally, I also find their conclusion that inbreeding depression is associated through the action of rare recessive variants to be completely unsupported by the analyses they have conducted.

To address this concern, we have done additional analyses which are reported in a new Supplementary Note 5. We acknowledge the reviewer’s comment below that FGRM is a better metric than FSNP and have modified this part of the study to focus on bivariate models of Trait \sim FROH + FGRM instead of Trait \sim FROH + FSNP_OutsideROH. This new bivariate model was performed in the entire ROHgen consortium but not previously reported. We also note that a recent study⁴⁵ found evidence for rare variants from similar analyses (Trait \sim FROH + FSNP), but provided less complete justification of their conclusions.

We have hopefully clarified (in Supplementary Notes 4, 5 and the Main Text) that the comparative magnitudes of Beta_FROH and Beta_FGRM, and particularly Beta_FGRM|FROH and Beta_FROH|FGRM, support the conclusion that the causal variants are in weak LD with common SNPs. We maintain that rare, recessive variants are therefore the most likely causal variants, but cannot exclude other variants in weak LD with the SNPs used to calculate FGRM (e.g. recurrent copy number or structural variants).

A recent study calls into question the use of ROH as a measure of inbreeding (Yengo et al. 2017 PNAS). The authors report that: “This attenuation is greatest in low homozygosity cohorts, suggesting that FROH is a better estimator of the true inbreeding coefficient (Fig 3a).” This is extremely weak evidence for this, if any evidence at all. This attenuation could be greatest in low homozygosity cohorts because of a wide-range of factors, including that FGRM is a better metric in unrelated, outbred samples for instance. This manuscript needs to fully justify it’s choice of metric by exploring through theory and simulation exactly why and when FGRM provides a worse measure of inbreeding than FROH because current evidence suggests that associations using FGRM are less biased, with results attenuated as compared to ROH and this is exactly what is found in this study. At the very very least the authors should present the FGRM results in the main text as the main results and highlight that many of their new and “surprising” findings completely disappear when using the alternative metric, which has been shown in extensive simulation study and theory to be less biased than the metric used here.

We have addressed this in Supplementary Notes 4 and 5. We do not agree, however, that many of our findings “completely disappear when using the alternative metric”. Indeed, 26 out of 32 remain significant at an experiment-wise level when associated with FGRM (p -value < 0.005) and all-but-one remain nominally significant (p -value < 0.05) (Supplementary Table 13). We believe this small difference in statistical power is because FROH has a stronger correlation with the excess homozygosity of the (rare) causal variants (Extended Data Fig. 16b).

ROH is more susceptible to stratification biases. Linear “dosage” responses, as the authors state, are also expected if ROH reflects the degree of population stratification and the covariance of environmental confounding. Thus, I find this to be no evidence that these associations do not reflect other underlying causes.

We agree with the reviewer that linear responses are also expected if ROH reflect the degree of population stratification or environmental confounding. The point we were trying to make was that it would be unlikely for ROH to reflect the degree of environmental cofounding across the *entire* range of FROH for which we observe a “dosed” response - all the way out to the offspring of first-degree relatives. It would be coincidental for the average environmental confounding associated with first-degree unions ($E(F)=0.25$) to be 4 greater times than the average environmental confounding associated with first-cousin (i.e. third degree) unions ($E(F)=0.0625$), and etc. This seems particularly true for reproductive or risk-related traits where the most likely confounder might be traditional, religious or conservative beliefs associated with cousin marriage. While it is easy to envisage that these confounders might be associated with intermediate FROH (~ 0.06) it is difficult to understand how these confounders would be four times more prevalent in the very unusual situation of first-degree unions. First-degree unions (incest) are of course not encouraged in any situation or culture of which we are aware. We have modified the Main Text to clarify our argument.

See also our reply to reviewer 2 below, which deals more generally with the issue of confounding.

FSNP (which is also a bad metric) outside of FROH tells one nothing of the minor allele frequency distribution of the directional dominance. No evidence in this manuscript is presented for an excess of rare-variant effects. One would have to repeat the entire analysis dividing the measure into MAF groups...so FGRM for rare versus common variants when both are fitted in the model. I note that rare variants are actually excluded from the ROH calling (which is fine), but it means that little can be said about the frequency distribution of the effects, leaving current statements as nothing more than conjecture.

FSNP_outsideROH was chosen in preference to FGRM_outside_ROH purely for practical reasons. We agree with the reviewer that FGRM_outside_ROH would be a better metric, but we were not able to calculate this within the constraints imposed by using established software that could be easily implemented by over 100 groups around the world (PLINK). However, we maintain that if causal variants were in LD with the SNPs used to calculate FSNP_outsideROH then FSNP_outsideROH would associate with the homozygosity of these causal variants and therefore also with Inbreeding Depression. The observation that for most traits FSNP_outsideROH does not associate with Inbreeding Depression indicates that the causal variants are not in strong LD with the SNPs used to calculate FSNP_outsideROH.

Although it was not practical to calculate FGRM_outside_ROH, we have added a new Supplementary Table 23 and Extended Data Figures 15a,b which present the meta-analysed results of bivariate models of Trait \sim FROH + FGRM. For all significant traits, the conditional effect of FROH is of greater magnitude than the conditional effect of FGRM, again indicating the causal variants are not in strong LD with the SNPs used to calculate FGRM. To support our interpretation of this result, in Supplementary Note 5 we have identified SNPs at specific MAF (0.01 to 0.5) and regressed the excess homozygosity of these SNPs on bivariate models of FROH + FGRM. We find that FGRM correlates more strongly with the excess homozygosity of common SNPs, while FROH correlates more strongly with rare SNPs. Since, in real data, the effects are preferentially associated with FROH (Supplementary Table 23 and Extended Data Figures 15a,b) we conclude that the causal variants are most likely rare.

In the replication studies, please present the replication effects and not the ratios as the ratio distort the interpretation, please let the reader have all of the information. This is in the supplementary but it's a question of unbiased presentation of the results. I note that the sibling pairs and the adopted individuals were not left out of the original meta-analysis, this would be nice to have unbiased replication of the effects.

We thank the reviewer for noting that the replication effects are available in the supplementary tables, and of course the replication p -values referred to in the text refer to these effect estimates. We presented replication ratios in Figs 4c,d to allow comparison across different trait units and indeed even across quantitative and binary traits. We can think of no better way to summarise the replications across a wide range of traits. In particular, we wanted to summarise the average degree

of replication across all traits in order to provide a global estimate of the degree of environmental confounding.

Line 443: linear mixed models are commonly used for binary traits under a liability threshold model to control for relatedness, so there is no reason to not control for family structure here.

We have added a sentence to the methods section to make clear that mixed modelling was used for all results obtained using a linear liability threshold model – i.e. all binary trait results except the effect estimates in Fig. 2b. We chose to use a full logistic model for the results of Fig. 2b, rather than a linear liability threshold model, only because of potential bias in the liability model in the case of extreme case: control ratios. In general, we see very good agreement between the two methods (Extended Data Fig. 8) except for self-declared infertility which has a prevalence of just 0.13% in these data.

The sibling analysis is very good and the method is very nice. Why don't you do it for the other metrics also? Also, please consider fitting the sibling IBD deviation from 0.5 of the pair to account for mendelian segregation as a whole and additive effects before testing for directional dominance.

We thank the reviewer for their appreciation of the siblings method. We believe it is important for the validity of future work to prove that observed effects are genetic inbreeding depression, and not artefacts of social confounding. Although we, and others, have presented 'circumstantial' evidence (consistency of effects, dosed response) to support a genetic hypothesis, the siblings method provides an absolute quantification of the genetic component. This realisation was made part-way through the current analyses, and we had not specifically targeted siblings in the original analysis plan. Despite this, two traits (overall health and reproductive success) reach replication significance and the "average" replication across all traits suggests an important genetic contribution to the others. We believe it will not be long until this method, applied to larger sibling datasets, will separate genetics from confounding for all traits of interest.

There are spousal pairs in the UK Biobank so why not test for assortment at ROH?

We thank the reviewer for this suggestion of interesting future work, but it seems beyond the scope of this current study.

Reviewer #2 (Remarks to the Author):

Re: Effects of autozygosity on a broad range of human phenotypes.

In the manuscript the author wish to determine a link between the lengths of autozygous regions in a subject to a number of phenotypic parameters which vary between those that are self-reported and physical measurements. The majority of the phenotypes also have a strong environmental component. Regarding the conclusions reached by authors, while I agree that increased inbreeding does have a negative impact on an individual, I have a major concern that the demonstrated effects on a number of the phenotypes are due societal pressure which limits a person's behaviour/options rather than an underlying mechanism where by a person's genetic makeup drives their behaviour.

A family's cultural background is a strong factor in the likelihood that they practice consanguineous unions and so the typical length of homozygous regions in family members. Similarly, a family's cultural background is a strong factor how they would perform/report on a large number of the parameters discussed in the paper. Consequently, I'm very concerned that the length of a person's homozygous regions is a marker for their cultural 'view point' and it is this that determines their phenotype/behaviour. For instance the age at which they first have sex, number of partners, amount of alcohol consumed, what they view as risk behaviour or how well they have aged, who they chose to have children with, etc. is strongly impacted by their cultural environment. Therefore, while the authors did adjust for ethnic origin and religion I think they need to more extensively explore any possible link between cultural 'view point' of the individuals and their phenotype. For instance Pakistani Muslims are not a homogeneous group with individuals originating from the north-west tribal regions more likely to be consanguineous and socially conservative than those from the large cities in the south, consequently population stratification may have a strong affect leading to the apparent linking of length of autozygous regions to a wide range of behaviours.

We thank the reviewer for their comments and wholeheartedly agree that confounding by cultural 'view point' is an important concern. As stated in reply to R1, above, we think that having established that these effects are primarily genetic is of great importance to the field of inbreeding research. For this reason, we have developed methods and devoted a substantial portion of the paper to disentangling the roles of genetics and environmental confounding (be that by cultural view point, or socioeconomic status, or indeed other unmeasured variables).

Evidence for the limited role of confounding, is summarised in five groupings below. That several of these are novel methods hopefully shows our appreciation of the importance of this concern.

- 1) **Constancy of effects in different cohorts.** The reviewer describes the case of Pakistani Muslims, and we agree that in any given cohort there is potential for associations between autozygosity and traits also affected by social confounders. However, the replication of effects in diverse situations (Extended Data Figs 1,2, Supplementary Tables 18,19) suggests that the observed effects are due to a more widespread cause. To take the specific examples mentioned by the reviewer (age at first sex, alcohol units, etc), many of these traits are indeed significantly associated with FROH in S&W Asian cohorts (Supplementary Table 18). However, they are also replicated, and highly significant, in the genetically British samples from UK Biobank. The cultural environment of these UK samples (self-declared white British and genetically white British by Principal Components) is likely to be very different to cultural situation in Pakistani and yet the effects are found to be consistent, as they are in other groups around the world.
- 2) **Proportional response to FROH.** As also written above in reply to reviewer 1, we found that the effects are proportional to FROH across the entire, wide, range of FROH observed. Although linear responses might also be expected when ROH reflect the degree of population stratification or environmental confounding, it would be unlikely for ROH to reflect the degree of environmental cofounding across the *entire* range of FROH for which we see a "dosed" response, all the way out to the offspring of first-degree relatives. It would be very coincidental for the average environmental confounding associated with first-degree unions ($E(F)=0.25$) to be 4 greater times than the average environmental confounding associated with first-cousin (i.e. third degree) unions ($E(F)=0.0625$), and etc. This seems

particularly true for reproductive or risk-related traits, where the most likely confounder might be traditional, religious, or conservative beliefs associated with cousin marriage. While it is easy to envisage that these confounders might be associated with intermediate FROH (~0.06) it is difficult to accept that these confounders would be four times more prevalent in the very unusual situation of first-degree unions. First-degree unions are of course not encouraged in any situation or culture of which we are aware. A linear response would, however, be the expectation of a genetic effect.

- 3) **Effects also observed for shorter ROH (1.5Mb to 5Mb).** (Extended Data Fig. 11a). Shorter ROH are prevalent in all cohorts due to inevitable co-ancestry deriving from finite ancestral population sizes. Unlike longer ROH, they are therefore less informative of the close parental relatedness which might be associated with cultural confounding. However, we find that, per unit length, shorter ROH have similar effects to longer ROH which is consistent with a genetic explanation where both long and short ROH make genetic variants homozygous through autozygosity.
- 4) **Effects replicated in a cohort of adopted samples.** (Fig. 4c). In UK Biobank 7153 samples report that they were adopted as a child. Although this is a small cohort in the context of this study, it has disproportionate statistical power due to a relatively large number of high FROH samples (16 out of 7153 have FROH > 0.18, compared to 33 out of 403,066 in the UKB British cohort). It seems likely that these individuals were put up for adoption, at least in part, due to the very close relatedness of their parents. We believe that adoption would have broken, or significantly weakened, any associations between high FROH and social factors also affecting traits. That we observe consistent effects in this cohort of adoptees suggests that the causes are primarily genetic.
- 5) **Effects replicated within full-sibling families.** Although we believe they are very unlikely, alternative explanations could be suggested to explain observations 1-4 above. However, as also written in response to Reviewer 1, we think this new method of comparing siblings is of real importance to the field, since it is entirely free of all possible confounding. Our analysis plan was not initially designed to target siblings, but nevertheless we have identified enough full-sibling pairs in a subset of the consortium cohorts to confirm two observed effects as genetic (reproductive success and overall health) and suggest a majority of the remainder are primarily genetic in origin. Due to random Mendelian segregation, children of the same two parents (full-siblings) inherit different haplotypes and therefore also different amounts of autozygosity. We find that the sibling with more autozygosity (FROH) is less likely to have children and reports poorer overall health than the sibling with less autozygosity. These two traits each reach replication significance in the sibling comparison (p-value < 0.0015). Although the other traits do not reach significant in the siblings-only analysis (height is very close), an average of all traits suggests they also are primarily genetic. Although siblings do not experience identical environments, there is no reason for the more autozygous sibling to experience systematically harsher environments — besides, possibly, the autozygosity itself. In this revised manuscript, we have added 2,626 siblings to this analysis which has strengthened the average replication in Fig. 4d.

I would also like to know a little more on how they filtered homozygous regions that spanned regions that are poorly genotyped in many genotype microarrays. For instances did they discount

homozygous runs that spanned the centromeres, particularly of chromosomes 17 and 19 where poor genotyping of regions flanking the centromere frequently suggests homozygosity. Since centromeres are not genotyped they don't contain heterozygous SNP which would terminate the extension of the apparent homozygous regions across the centromere. This can result in the false reporting of homozygous runs of over 5Mb per chromosome.

Among the ROH calling parameters we used (See Methods) two are directly relevant to this question: homozyg-gap 1000 and homozyg-density 50. The former prevents any ROH with a gap of >1000kb between adjacent SNPs and the latter requires there to be an average of at least one SNP every 50kb in each ROH. These parameters appear to be sufficient to prevent excess ROH calling at centromeres:

Although not reported in the manuscript, we calculated the average ROH frequency in each of 1000 3MB-wide windows for all cohorts. Averaging over all cohorts, the median ROH frequency is 0.39%, although 15% of windows contain an ROH in more than 1% of individuals. These high homozygosity windows are often called ROH islands and occur where a population contains common long (>1.5Mb) haplotypes. The windows covering the centromeres specifically mentioned (19:24-27MB and 17:24-27MB) have average autozygosities of 0.23% and 0.49% and thus fall respectively in the 9th and 61st percentile of all windows, suggesting there is little if any false reporting of ROH in these regions.

Reviewer #3 (Remarks to the Author):

This study examined the relationship between individual-level autozygosity and a wide range of complex traits, reporting that autozygosity is associated with 32 of 100 traits analyzed. Even though this result is not surprising, it is important because the analysis was based on a very large sample and a large number of traits, and it has carefully explored potential confounding factors by using adopted individuals and sib pairs. What is particularly impressive is that the results are consistent across diverse ancestral groups and across several demographic scenarios that could increase autozygosity.

We thank the reviewer for their positive comments, and in particular for their appreciation of the consideration given to potential confounding.

I have only a few minor comments.

1. The genetics concepts can be defined more systematically. The study used long ROH regions as a surrogate of autozygosity of recent origin. More precisely, the study used genotype data to identify unusually long regions of ROH, which cannot be explained by the expected length distribution of ROH runs in an ideal population, with a large size and no inbreeding in the recent and mid-distance past. Operationally the study used 1.5 Mb as the cutoff, and used the total length of such long-ROH segments in an individual to calculate his/her genomic fraction accounted for by these regions. In contrast, the genomewide heterozygosity rate contains both the long-ROH fraction calculated above and the excess amount of homozygous genotypes outside of long-ROH regions. I think the authors

did the right analyses, but hope it can be explained even more clearly. In particular, whether the second measure is based on each population's own allele frequencies is hard to find, and I assume it is and it is somewhere in the Method section. The discrepancy between these two measures reflects the relative contribution of autozygosity from different time depths; that is, how much is from very recent inbreeding events and how much is from more distant events, including the situation loosely called small effective population size.

We agree with the reviewer's description of the genetic concepts and study objectives. The "excess amount of homozygous genotypes outside of long-ROH regions" is indeed based on each population's own allele frequencies and we have modified the methods to make this clear. As regards making the interpretation of these analyses clearer, we have now included a new description of bivariate models of FROH and FGRM. The interpretation of this new analysis is explored in Supplementary Note 5 where we show that the observed effects are associated with rare variants (made homozygous by recent inbreeding) but not with common variants (made homozygous by both recent and distant inbreeding).

2. Some other technical details are important enough that I think they can be brought forth, from Method to the main section. The Result mentioned "231 homogeneous sub-cohort". Here "homogeneous" probably meant there is no discernible within-cohort stratification, but each could have its own level of autozygosity. It is not immediately clear if homogeneity is based on a formal test, such as patterns of F_{st} or HWE test statistics over all the loci. F_{IS} calculation is affected by the genotyping panel (whether it contains mainly common or rare variants); so it seems that we need to be told more about potential caveats if different SNP arrays were used. The study nicely shows that the comparison of the two autozygosity measures: long-ROH vs. all-the-rest, can divide the cohorts into at least three demographic scenarios. However, one of them, large F_{IS} and small F-ROH, is for non-homogeneous cohorts.

The homogeneity of each cohort was not formally tested, beyond instructions to each analyst to ensure that each cohort contained a single ethnicity, genotyped on a single SNP array. By "homogeneous" we wanted to convey that the cohorts contain a single ethnicity, but we agree that the large F_{IS} , small FROH cohorts are not genetically homogeneous since they contain admixed ethnicities (i.e. Hispanic, Caribbean, etc.). Although these are recognised ethnicities, varying ancestral proportions may result in genetic population structure. We have replaced "homogeneous" with "single-ethnicity" or "uniform" where applicable.

3. A reader who only skim through the main text may still wonder if the effect size – the slope in the regression between the trait and FROH - is highly variable among the populations or ancestral groups. Maybe I missed it. In any case this point can be provided more centrally.

We have attempted to make this point clearer at the bottom of page 12 and in the Discussion (page 17), as well as Supplementary Tables 18,19 and Extended Data Figures 1,2.

4. Age effect has been "removed" by being treated as a covariate in the model. I wonder if the effect of autozygosity has an interaction effect with age, for some of the traits.

We thank the reviewer for this interesting suggestion, but given the consortial nature of the study is not possible retrospectively. We will consider including this in future iterations of the consortium.

5. Some phrases can be defined more clearly. "genetically correlated" is one. "true inbreeding coefficient" is problematic because different measures emphasize different time depth, and the field has been handicapped for so long by dichotomizing the concept into very recent (consanguinity) and very distant (small population size). In the context of this work, FROH captures a greater span of the genome than F_{IS} . Whether 1% in FROH is truly more "risky" than 1% in F_{IS} is a profound question. Technically it's difficult to calculate a score like F_{IS} that is specific for a defined model, such as co-ancestry in 10-200 generations-but-not-more-recent. Here the question is related to harder questions such as, is a rare recessive variant more impactful when it is contained in a long ROH than in a shorter ROH? I think this paper brought many exciting questions for the future.

We agree that there are many exciting questions to be answered! By "true inbreeding coefficient" we understand "inbreeding coefficient (or excess homozygosity) at the causal loci" and have modified the phrase accordingly to "excess homozygosity at the causal loci". We have attempted to define this precisely as F_{QTL} in Supplementary Note 5. Since these causal loci are not yet identified we concur that dichotomising inbreeding into very recent (consanguinity) and very distant (small population size) is not particularly helpful. In Supplementary Note 5 we have explored the question of whether 1% FROH is more impactful than 1% FGRM and conclude that FROH is more strongly associated with the effects observed. We show that this implies the causal variants are most likely rare, not common.

Reviewers' Comments:

Reviewer #1:

Remarks to the Author:

The authors should be commended for conducting a tremendous amount of research in order to address my concerns and that of the other reviewers.

I feel that this work is well deserving of publication in Nature Communications as this discourse regarding which metrics to use (i.e. what is the best way to capture inbreeding from SNP data and estimate its effects on complex traits), should be seen by the research community, and the work conducted here contributes considerably to this debate.

I do not agree fully with everything that was done, but to ask for more work at this stage would be well beyond the scope of one publication. In particular, simulation in real data is difficult as one does not know the truth, and I feel that there remain some theoretical aspects to explore. However, I feel that what has been conducted is accurately presented and rigorous. I think this manuscript will lead to many follow-up publications exploring the theory in more detail, I think it is a valuable contribution to the literature that will be cited many times, and I feel that this is an important question for the field of genetics.

Reviewer #2:

Remarks to the Author:

I still think that the total length of autozygous regions in consanguineous and inbred populations is a strong marker for cultural conservatism and so likely to have a strong influence over how individuals behave. The authors mention individuals who are the result of incest and while first degree unions are not the norm for any grouping I'm aware of, I think that any child born in these circumstances will probably live in a very unusual situation and so likely will not follow any social trend and so be different as a whole from the general population.

Regarding analysing adopted children: It is my understanding that social services in Britain spend a considerable amount of effort to match the adopting parents to the child's biological parents on parameters such as ethnicity, religion and cultural background so the link between high FROH and social factors may not be totally broken as the authors suggest. Also, the basic thrust of the paper suggests that they have analysed a very large cohort of individuals, but support this assertion based on a very small subsample of the overall cohort.

While I generally think it's unfair for a reviewer make new points halfway through the review process, it did strike me after submitting my initial comments is that if this data had been collected in 1940, the spectrum of autozygosity would be very similar to what it is today, but the answers to all the questions of risk taking, sexual activity etc. would be very different, with the answers for white Europeans more similar to Asians than it is today. Consequently, I tend to think that any effect you would see is predominantly due to cultural stratification and not genetics which is the reason that physical traits like height and weight aren't statistically significant, but traits affected by social pressure are.

Reviewer #3:

Remarks to the Author:

I am satisfied with the revision.

I hope the authors can incorporate this reference:

<https://www.ncbi.nlm.nih.gov/pubmed/26940866>

To a lesser extent: <https://www.ncbi.nlm.nih.gov/pubmed/29691392>

<https://www.ncbi.nlm.nih.gov/pubmed/28406212>

I appreciate the new Supplementary Notes 4-5. A good example of quantitative analysis of mutation load (individual fitness) is in <https://www.ncbi.nlm.nih.gov/pubmed/25279984>, where variants were stratified by frequency.

For a broader perspective: <https://www.ncbi.nlm.nih.gov/pubmed/29325102>

We again thank all the reviewers for their time and expertise in considering our revised manuscript

REVIEWERS' COMMENTS:

Reviewer #1 (Remarks to the Author):

The authors should be commended for conducting a tremendous amount of research in order to address my concerns and that of the other reviewers.

I feel that this work is well deserving of publication in Nature Communications as this discourse regarding which metrics to use (i.e. what is the best way to capture inbreeding from SNP data and estimate its effects on complex traits), should be seen by the research community, and the work conducted here contributes considerably to this debate.

I do not agree fully with everything that was done, but to ask for more work at this stage would be well beyond the scope of one publication. In particular, simulation in real data is difficult as one does not know the truth, and I feel that there remain some theoretical aspects to explore. However, I feel that what has been conducted is accurately presented and rigorous. I think this manuscript will lead to many follow-up publications exploring the theory in more detail, I think it is a valuable contribution to the literature that will be cited many times, and I feel that this is an important question for the field of genetics.

We specifically thank this reviewer for their generous comments and consideration given to the new analyses. We agree there are many exciting questions still to be answered in this field and look forward to all follow-up publications.

Reviewer #2 (Remarks to the Author):

I still think that the total length of autozygous regions in consanguineous and inbred populations is a strong marker for cultural conservatism and so likely to have a strong influence over how individuals behave. The authors mention individuals who are the result of incest and while first degree unions are not the norm for any grouping I'm aware of, I think that any child born in these circumstances will probably live in a very unusual situation and so likely will not follow any social trend and so be different as a whole from the general population.

Regarding analysing adopted children: It is my understanding that social services in Britain spend a considerable amount of effort to match the adopting parents to the child's biological parents on parameters such as ethnicity, religion and cultural background so the link between high FROH and social factors may not be totally broken as the authors suggest. Also, the basic thrust of the paper suggests that they have analysed a very large cohort of individuals, but support this assertion based on a very small subsample of the overall cohort.

While I generally think it's unfair for a reviewer to make new points halfway through the review process, it did strike me after submitting my initial comments is that if this data had been collected in 1940, the spectrum of autozygosity would be very similar to what it is today, but the answers to all the questions of risk taking, sexual activity etc. would be very different, with the answers for white Europeans more similar to Asians than it is today. Consequently, I tend to think that any effect you would see is predominantly due to cultural stratification and not genetics which is the reason

that physical traits like height and weight aren't statistically significant, but traits affected by social pressure are.

We agree that because F_{ROH} is inevitably confounded with cultural factors there are likely to be non-genetic associations, of some magnitude, with all traits. It is for this reason that we draw particular attention to the within-siblings analysis, which quantifies the purely genetic component. The current sample size confirms that two of the traits have a significant genetic component, and shows that, overall, only about 20% of the observed effects are attributable to non-genetic factors. This new within-siblings method will also allow future, larger studies to more precisely quantify for every trait, the relative contributions of genetic and cultural effects on F_{ROH} .

We have added a new paragraph to the discussion which deals with the question of cultural confounding in much more detail than previously.

Reviewer #3 (Remarks to the Author):

I am satisfied with the revision.

I hope the authors can incorporate this reference:

<https://www.ncbi.nlm.nih.gov/pubmed/26940866>

To a lesser extent: <https://www.ncbi.nlm.nih.gov/pubmed/29691392>

<https://www.ncbi.nlm.nih.gov/pubmed/28406212>

I appreciate the new Supplementary Notes 4-5. A good example of quantitative analysis of mutation load (individual fitness) is in <https://www.ncbi.nlm.nih.gov/pubmed/25279984>, where variants were stratified by frequency.

For a broader perspective: <https://www.ncbi.nlm.nih.gov/pubmed/29325102>

We appreciate these references and have added Narasimhan *et al.* and Saleheen *et al.* to the manuscript.